# Long-term high-grain diet altered the ruminal pH, fermentation, and composition and functions of the rumen bacterial community, leading to enhanced lactic acid production in Japanese Black beef cattle during fattening

**Toru Ogata[1,2], Hiroki Makino[2], Naoki Ishizuka[2], Eiji Iwamoto[3], Tatsunori Masaki[3], Kentaro Ikuta[4], Yo-Han Kim⊙[2]*, Shigeru Sato⊙[1,2]***

**1** United Graduate School of Veterinary Sciences, Gifu University, Gifu, Japan, **2** Cooperative Department of Veterinary Medicine, Faculty of Agriculture, Iwate University, Morioka, Iwate, Japan, **3** Hyogo Prefectural Technology Center of Agriculture, Forestry and Fisheries, Hyogo, Japan, **4** Awaji Agricultural Technology Center, Minami-Awaji, Hyogo, Japan

* haneey@iwate-u.ac.jp (YHK); sshigeru@iwate-u.ac.jp (SS)

**Data Availability Statement:** The sequence data were deposited into the Sequence Read Archive of

## Abstract

To increase intramuscular fat accumulation, Japanese Black cattle are commonly fed a high-grain diet from 10 to 30 months of age although it can result in the abnormal accumulation of organic acids in the rumen. We explored the effect of long-term high-concentrate diet feeding on ruminal pH and fermentation, and its effect on the rumen bacterial community in Japanese Black beef cattle during a 20-month fattening period. Nine castrated and fistulated Japanese Black beef cattle were housed with free access to food and water throughout the study period (10–30 months of age). The fattening stages included Early, Middle, and Late stages (10–14, 15–22, and 23–30 months of age, respectively). Cattle were fed high-concentrate diets for the experimental cattle during fattening. The body weight of the cattle was 439 ± 7.6, 561 ± 11.6, and 712 ± 18.5 kg (mean ± SE) during the Early, Middle, and Late stages, respectively. Ruminal pH was measured continuously during the final 7 days of each stage, and rumen fluid and blood samples were collected on day 4 (fourth day during the final 7 days of the pH measurements). The 24-h mean ruminal pH during the Late stage was significantly lower than that during the Early stage. Total volatile fatty acid (VFA) during the Late stage was significantly lower than during the Early and Middle stages, but no changes were noted in individual VFA components. The lactic acid concentration during the Late stage was significantly higher than that during the Early and Middle stages. The bacterial richness indices decreased significantly during the Late stage in accordance with the 24-h mean ruminal pH. Among the 35 bacterial operational taxonomic units (OTUs) shared by all samples, the relative abundances of OTU8 (Family *Ruminococcaceae*) and OTU26 (Genus *Butyrivibrio*) were positively correlated with the 24-h mean ruminal pH. Total VFA concentration was negatively correlated with OTU167 (Genus *Intestinimonas*), and lactic acid concentration was correlated positively with OTU167 and OTU238 (Family *Lachnospiraceae*). These results suggested that long-term high-grain diet feeding gradually lowers

the National Center for Biotechnology Information and can be accessed via SRA accession number PRJNA548210 (https://submit.ncbi.nlm.nih.gov/subs/sra/).

**Funding:** The authors received no specific funding for this work.

**Competing interests:** The authors have declared that no competing interests exist.

ruminal pH and total VFA production during the Late fattening stage. However, the ruminal bacterial community adapted to feeding management and the lower pH during the Late stage by preserving their diversity or altering their richness, composition, and function, to enhance lactic acid production in Japanese Black beef cattle.

## Introduction

A high-grain based diet is essential for beef and dairy cattle, to maximize growth, productivity, and high-quality meat or milk. However, highly fermentable carbohydrate feeding can result in the accumulation of organic acids in the rumen, such as volatile fatty acids (VFAs) and lactic acid [1, 2]. As a result, ruminal pH decreases; subacute ruminal acidosis (SARA) and ruminal acidosis (RA) are defined by ruminal pHs values of $\leq 5.6$ and below, respectively [2]. The production of organic acids by microbes, and their removal or neutralization by the gastrointestinal tract, constitutes a well-balanced regulatory system in the rumen [1]. The ruminal bacterial community and ruminal pH can adapt to and influence each other [3], and the effects of short- (days) and mid-term (weeks) SARA and RA challenges have been explored previously [3, 4, 5, 6]. In general, the ruminal bacterial community has similar proportions between the phyla *Firmicutes* and *Bacteroidetes* under a high-forage diet with higher ruminal pH [3, 7, 8]. However, grain-based SARA challenge induces death or lysis of Gram-negative bacteria, such as *Bacteroidetes* and *Proteobacteria*, and eventually the proportion of *Firmicutes* increases with severely low ruminal pH [3, 7, 8].

Japanese Black cattle are characterized by the ability to deposit a large amount of intramuscular fat [9]. The fattening of Japanese Black cattle typically begins at about 10 months and is completed by about 30 months of age. Although hypovitaminosis A may be related to the occurrence of hepatic disorders, the fattening cattle are generally fed high-grain, low vitamin A-containing diets to induce greater intramuscular fat deposition, leading to highly marbled meat during the fattening period [10]. However, limited information is available on the effects of a rumen environment on Japanese Black cattle fattening, in terms of the ruminal pH, fermentation, and bacterial communities.

Therefore, we explored the effects of the ruminal pH, bacterial community and fermentation characteristics on different ages of Japanese Black beef cattle fed a long-term (20-month) high-grain diet. In addition, these findings increase our understanding of the ruminal pH, fermentation, and bacterial community as an adaptation to long-term high-grain diet, thereby contributing to the understanding of their relationships in Japanese Black beef cattle.

## Materials and methods

An experimental protocol was approved by the Iwate University Laboratory Animal Care and Use Committee (A201720; Morioka, Japan), and all animal experiments were conducted following the animal experiment policy of Hyogo Prefectural Technology Center for Agriculture, Forestry and Fisheries (Hyogo Prefecture, Japan).

### Animals and experimental design

A total of nine castrated (at age 5–6 months) and subsequently fistulated (at age 12 months under local anesthesia) Japanese Black beef cattle were housed with free access to food and water throughout the study period (10–30 months of age). The fattening stages included

**Table 1. Body weight, dietary composition, and chemical analysis of diets in Japanese Black beef cattle during the Early, Middle, and Late fattening stages.**

| Items | Stage[1] | | | SEM |
|---|---|---|---|---|
| | **Early** | **Middle** | **Late** | |
| Body weight (kg) | 439.1[a] | 561.8[b] | 712.4[c] | 12.6 |
| Daily intake amount[2] (kg) | | | | |
| Concentrate[3] | 6.0[a] | 7.6[b] | 6.1[a] | 0.32 |
| Rice straw | 2.1[a] | 1.1[b] | 1.0[b] | 0.13 |
| Nutrient adequacy rate[4] (%) | | | | |
| DM[5] | 88.7[a] | 96.1[a] | 75.4[b] | 3.48 |
| TDN[6] | 91.2[a] | 102.4[a] | 74.2[b] | 3.85 |
| NDF[7] | 43.9[a] | 36.8[b] | 31.5[c] | 0.54 |

[a,b,c]Mean within a row, different superscripts significantly differ ($P < 0.05$)

[1]The age of cattle in the Early, Middle, and Late stages were 14, 21, and 29 months, respectively.

[2]Organic matter basis

[3]The concentrate diet composed of barely, steam-flaked corn, wheat bran, and soybean meal and contains 71.2% total digestible nutrient (TDN) and 15.7% crude protein (CP), 72.2% TDN and 13.9% CP, and 72.8% TDN and 12.0% CP during the Early, Middle, and Late stage, respectively.

[4]Nutrient adequacy rate was based on the nutrient requirement of Japanese Feeding Standard for Beef Cattle [13], with an expected daily weight gain of 0.8, 0.65, and 0.7 kg during the Early, Middle, and Late stages, respectively.

[5]DM = dry matter

[6]TDN = total digestible nutrients

[7]NDF = neutral detergent fiber.

Early, Middle, and Late stages (10–14, 15–22, and 23–30 months of age, respectively) according to general agreement in Japan [10, 11]. The concentrate diet and rice straw were given a calculated amount for daily gain of 0.8 kg/day during the Early stage and *ad libitum* during the Middle and Late fattening stages (Table 1). The concentrate diet was composed of barely, steam-flaked corn, wheat bran, and soybean meal and contains 71.2% total digestible nutrient (TDN) and 15.7% crude protein (CP), 72.2% TDN and 13.9% CP, and 72.8% TDN and 12.0% CP during the Early, Middle, and Late stage, respectively. Feed refusal rate of concentrate and forage diet were 12.6 and 12.2%, respectively, during the Early stage. The forage-to-concentrate ratio was 26:74, 13:87, and 14:86 during the Early, Middle, and Late stages, respectively. The mean ± SE body weight of the cattle was 335 ± 4.4, 439 ± 7.6, 562 ± 11.6, and 712 ± 18.5 kg on prior to the experiment (10 months of age), and Early (14 months of age), Middle (21 months of age), and Late (29 months of age) fattening stage sampling days, respectively. The forage diet was supplied daily in two equal portions at 0930 and 1530 h, and concentrate diet was supplied 1 h after a forage diet feeding to maximize forage diet intake and to prevent excessive consumption of concentrate diet during the Early stage. Abnormalities of body condition (body temperature, appetite, hydration, and defecation) were observed daily throughout the study period. Daily intake amounts of concentrate and forage were recorded daily during the final 7 days (days 1–7) of the Early, Middle, and Late fattening stages. The body weight, intake amount, and chemical composition of the Early, Middle, and Late fattening stage diets are shown in Table 1. Chemical composition of the diets was analyzed according to the official method analysis of the Association of Official Analytical Chemists (AOAC) that registered in the Official Method Feed Analysis of Japan [12]. The adequacy rate of diet was calculated based on the nutrient requirement of Japanese Feeding Standard for Beef cattle [13].

## Sampling and measurements

Ruminal pH was measured continuously every 10 minutes during the final 7 days (days 1–7) of the Early, Middle, and Late fattening stages using a radio transmission system (YCOW-S; DKK-TOA, Yamagata, Japan), as described previously [14]. A pH sensor was placed in the ventral sac of the rumen through the rumen fistula. Calibration was performed at standard pH values of 4–7, before and after obtaining data in each fattening stage; no change in pH was observed during calibration. Rumen fluid samples were collected on day 4 (fourth day during the final 7 days of the pH measurements) during the Early, Middle, and Late stages, for analysis of the bacterial community, total VFA, individual VFAs, lactic acid concentration, and lipo-polysaccharide (LPS) activity. The collected samples were immediately filtered through two layers of cheesecloth and stored at −80˚C until use.

For the VFA analyses, 1 mL of 25% $HO_3P$ in $3 N H_2SO_4$ was added to 5 mL of rumen fluid. Total VFA and individual VFAs (i.e., acetic acid, propionic acid, and butyric acid) were separated and quantified by gas chromatography (GC-2014; Shimadzu, Kyoto, Japan) using a packed glass column (Thermon-3000; 3%) with a Shimalite TPA 60–80 mesh support (Shinwa Chemical Industries Ltd., Kyoto, Japan). For lactic acid analyses, fluid samples were centrifuged at $2,000 \times g$ for 15 min at 4˚C, and the concentration of lactic acid in the supernatant was determined using a commercially available kit (F-kit [d-lactate/l-lactate]; J.K. International Co., Tokyo, Japan). To measure the $NH_3$-N level, fluid samples were analyzed using the steam distillation method with an $NH_3$-N analyzer (Kjeltec Auto Sampler System 1035 Analyzer; Tecator Inc., Höganäs, Sweden). To measure ruminal LPS activity, rumen fluid samples were centrifuged at $11,000 \times g$ for 15 min at 4˚C, and supernatant LPS activity was assayed using a kinetic *Limulus* amebocyte lysate assay (Pyrochrome with Glucashield; Seikagaku Corporation, Tokyo, Japan). Details of the sample preparation and method validation procedures have been described previously [15].

## DNA isolation

Total bacterial DNA was extracted from rumen fluid samples, as described previously [16]. Briefly, samples were incubated with 750 μg/mL lysozyme (Sigma-Aldrich Co., St. Louis, MO, USA) at 37˚C for 90 min. This was followed by the addition of 10 μL of purified achromopepti-dase (Wako Pure Chemical Industries Ltd., Osaka, Japan) at a concentration of 10,000 U/mL, and the resulting mixture was incubated at 37˚C for 30 min. This suspension was treated with 60 μL of 1% sodium dodecyl sulfate and 1 mg/mL proteinase K (Merck Japan Ltd., Tokyo, Japan) and incubated at 55˚C for 5 min. Lysate was treated three times with phenol/chloro-form/isoamyl alcohol (25:24:1) (Wako Pure Chemical Industries Ltd.) and chloroform (Life Technologies Japan Ltd., Tokyo, Japan). DNA was precipitated with 5 M NaCl and 100% etha-nol, followed by centrifugation at $21,900 \times g$ for 15 min at 4˚C. The DNA pellet was rinsed with 70% ethanol, dried, and dissolved in Tris-hydrochloride buffer. Purified DNA was quan-tified using a Biospec-nano spectrophotometer (Shimadzu Biotech, Kyoto, Japan) and stored at −80˚C until further analyses.

## Library preparation and DNA sequencing

Sequencing libraries preparation was performed according to the Illumina 16S Metagenomic Sequencing Library preparation guide (2013) [17]. Bacterial 16S rRNA gene was amplified using barcoded universal primers 341F (5′–CCTACGGGNGGCWGCAG–3′) and 805R (5′–GACTACHVGGGTATCTAATCC–3′) spanning the V3–V4 hyper variable region [18]. Polymer-ase chain reaction (PCR) was performed on a 25 μL mixture containing 12.5 μL of $2 \times$ KAPA HiFi HotStart ReadyMix (Kapa Biosystems Ltd., UK), 5 μL of each primer (1 μM), and 2.5 μL

of template DNA (10 ng/μL). The thermal cycling conditions were 95˚C for 3 min, followed by 25 cycles at 95˚C for 30 s, 55˚C for 30 s, and 72˚C for 30 s, and a final extension at 72˚C for 5 min. Amplicons were purified using AMPure XP beads (Beckman Coulter, High Wycombe, UK) according to the manufacturer's instructions. Libraries were constructed by ligating sequencing adapters and indices onto purified PCR products using the Nextera XT Sample Preparation Kit (Illumina, San Diego, CA, USA) according to the manufacturer's instructions. Paired-end sequencing (2 × 150 bp) was conducted on the Illumina MiSeq platform according to standard protocols. The sequence data were deposited into the Sequence Read Archive of the National Center for Biotechnology Information and can be accessed via SRA accession number PRJNA548210 (https://submit.ncbi.nlm.nih.gov/subs/sra/).

## Sequencing data analyses

All sequencing reads were processed using the MOTHUR program (version 1.41.1; University of Michigan; http://www.mothur.org/wiki/; [19]), following the standard operating procedure for MiSeq (https://mothur.org/wiki/MiSeq_SOP; [20]) with minor modifications. To obtain a non-redundant set of sequences, unique sequences were identified and aligned against the SILVA reference database (SSURef release 128; [21]); then, candidate sequences were screened and filtered, unique sequences were determined, candidate sequences were pre-clustered to eliminate outliers, chimeras were removed using the "chimera.vsearch" command, and sequence comparisons were performed using the Mothur Ribosomal Database Project (RDP) training set (version 16). Sequences identified as being of eukaryotic origin were removed and a distance matrix was generated from the remaining sequences. Sequences were clustered and classified into operational taxonomic units (OTUs) using a cutoff value of 97% similarity. All samples were standardized by random subsampling (6,741 sequences/sample) using the "sub. sample" command, resulting in the elimination of two samples from the Middle and Late stages, which were then subjected to further analysis. The OTU values and rarefaction curves for each group were analyzed using the "rarefaction.single" command according to the 97% similarity cutoff. The "summary.single" command was used to analyze the OTU, Chao1, abundance-based coverage estimator (ACE) richness indices and Shannon, Simpson, and Heip diversity indices.

Representative sequences for each OTU were determined using the "get.oturep" command, and sequence comparisons were performed using the BLASTn program (https://blast.ncbi. nlm.nih.gov/Blast.cgi) against a 16S ribosomal RNA sequence database (Bacteria and Archaea; May 2019). Representative sequences and tabulated raw count data were submitted to the piphillin website (http://piphillin.secondgenome.com/; [22]). For the analysis of functional categories, a sequence identity cutoff of 97% was applied, and metagenomic functions were assigned using the Kyoto Encyclopedia of Genes and Genomes (KEGG) database (October 2018).

## Blood sampling and plasma metabolite profiles

Blood samples were collected on day 4 (fourth day during the final 7 days of the pH measurements), with rumen fluid samples from the jugular vein collected into 10 mL evacuated serum-separating tubes and tubes containing heparin (BD Vacutainer, Franklin Lakes, NJ, USA). Samples were immediately centrifuged (1,500 × g, 15 min, 4˚C) to separate the serum and plasma, and then preserved at –80˚C until analyses. Total protein (TP), blood urea nitrogen (BUN), total cholesterol (T-CHO), aspartate transaminase (AST), γ-glutamyl transpeptidase (GGT), and calcium (Ca) were measured using an automated biochemistry analyzer (Accute, Toshiba Ltd., Tokyo, Japan). Concentrations of retinol (vitamin A), α-tocopherol (vitamin E),

and β-carotene were analyzed by high-performance liquid chromatography according to Kata-moto et al. [23]. The plasma concentration of lipopolysaccharide-binding protein (LBP) was measured using a commercially available kit (HK503; HyCult Biotechnology, Uden, The Neth-erlands) according to Takemura et al. [24].

### Statistical analyses

The normality of the data distribution was assessed using the Shapiro-Wilk test. Significant differences in ruminal pH, duration of time where pH < 5.6 and < 5.8, area under curve (AUC) values for pH < 5.6 and < 5.8, VFAs, lactic acid concentration, LPS activity, and blood metabolites among the Early, Middle, and Late stages were evaluated using paired *t*-test for normal variables and the Wilcoxon rank sum test for non-normal variables. Significant differ-ences in the relative abundances of bacterial phyla, genera, OTUs, and bacterial richness and diversity indices among the Early, Middle, and Late stages were evaluated using the unpaired t-test for normal variables and Mann–Whitney U test for non-normal variables. Principal component analysis (PCA) plots were constructed using the R package ggbiplot (R software version 3.3.2; R Foundation for Statistical Computing, Vienna, Austria), including the 24-h mean ruminal pH, duration of time where pH < 5.6 and < 5.8, and AUC values for pH < 5.6 and < 5.8, and non-metric multidimensional scaling (NMDS) plots were constructed using the R package ggplot, including the OTUs and KEGG pathway categories. Pearson's correla-tion coefficients (r) were calculated between the rumen parameters (24-h mean, minimum, and maximum ruminal pH, duration of time where pH < 5.6 and < 5.8, total VFA and lactic acid concentrations, proportions of individual VFAs, LPS activity, and peripheral blood LBP concentration) and OTUs. A heatmap was constructed using Prism software (version 8.10; GraphPad Software Inc., La Jolla, CA, USA) based on the Pearson correlation data. All numer-ical data were also analyzed using Prism. A *P*-value < 0.05 was considered to indicate a signifi-cant difference, while P < 0.10 was taken as a trend towards significance.

## Results

### Body weight and dietary intake

No adverse health condition throughout the study period and effects of ruminal cannulation after surgery were observed for any of the cattle. The body weight of the Japanese Black cattle increased gradually but significantly across the Early, Middle, and Late fattening stages (*P* < 0.05; Table 1). The concentrate diet intake during the Middle stage was significantly higher than that during the Early and Late stages (*P* < 0.05), and forage consumption during the Early stage was significantly higher than that during the Middle and Late stages (*P* < 0.05). Nutrient adequacy rates of dry matter (DM) and TDN, calculated based on the total consump-tion amounts of concentrate diet and forage, during the Late stage were significantly lower than during the Early and Middle stages (*P* < 0.05), while that of neutral detergent fiber (NDF) was significantly higher during the Middle stage compared with the Early stage, and during the Late stage versus the Early and Middle stages (*P* < 0.05; Table 1).

### Ruminal pH, VFAs, and blood metabolites

The 24-h ruminal pH data were summarized as minimum, mean, and maximum pH values, duration of time where pH < 5.6 and < 5.8, and AUC values for pH < 5.6 and < 5.8 (Table 2). The minimum and mean ruminal pH during the Late stage were significantly lower than those during the Early stage (*P* < 0.05). In accordance with the 24-h mean ruminal pH, the duration of time where pH < 5.6 was significantly longer, and the AUC value for pH < 5.6 was

**Table 2. The 24-h mean ruminal pH, duration of time, and area under curve (for pH <5.6 and 5.8) in Japanese Black beef cattle during the Early, Middle, and Late fattening stages.**

| Item | Stage | | | SEM |
|---|---|---|---|---|
| | **Early** | **Middle** | **Late** | |
| 24-h mean ruminal pH | | | | |
| Minimum | 5.43[a] | 5.30[ab] | 4.98[b] | 0.10 |
| Mean | 6.22[a] | 6.06[ab] | 5.73[b] | 0.03 |
| Maximum | 6.79 | 6.76 | 6.69 | 0.09 |
| Duration of ruminal pH (min/d) | | | | |
| pH <5.6 | 139[a] | 287[a] | 688[b] | 105 |
| pH <5.8 | 226[a] | 460[ab] | 802[b] | 110 |
| Area under curve (pH × min/d) | | | | |
| pH <5.6 | 4.29[a] | 7.68[ab] | 13.7[b] | 5.17 |
| pH <5.8 | 5.24[a] | 12.7[ab] | 24.0[b] | 1.51 |

[a,b]Mean within a row, different superscripts significantly differ ($P < 0.05$)

significantly higher, during the Late stage than the Early stage (p < 0.05). In addition, the duration of time where pH < 5.8 during the Late stage was also significantly longer than during the Early and Middle stages ($P < 0.05$). Diurnal changes in the 10-minute mean ruminal pH were shown for the Early, Middle, and Late stages (Fig 1), and a gradual decrease in ruminal pH during the latter fattening stages was seen.

Total VFA concentration during the Late stage was significantly lower than during the Early and Middle stages ($P < 0.05$; Table 3). The ruminal acetic acid-to-propionic acid ratio

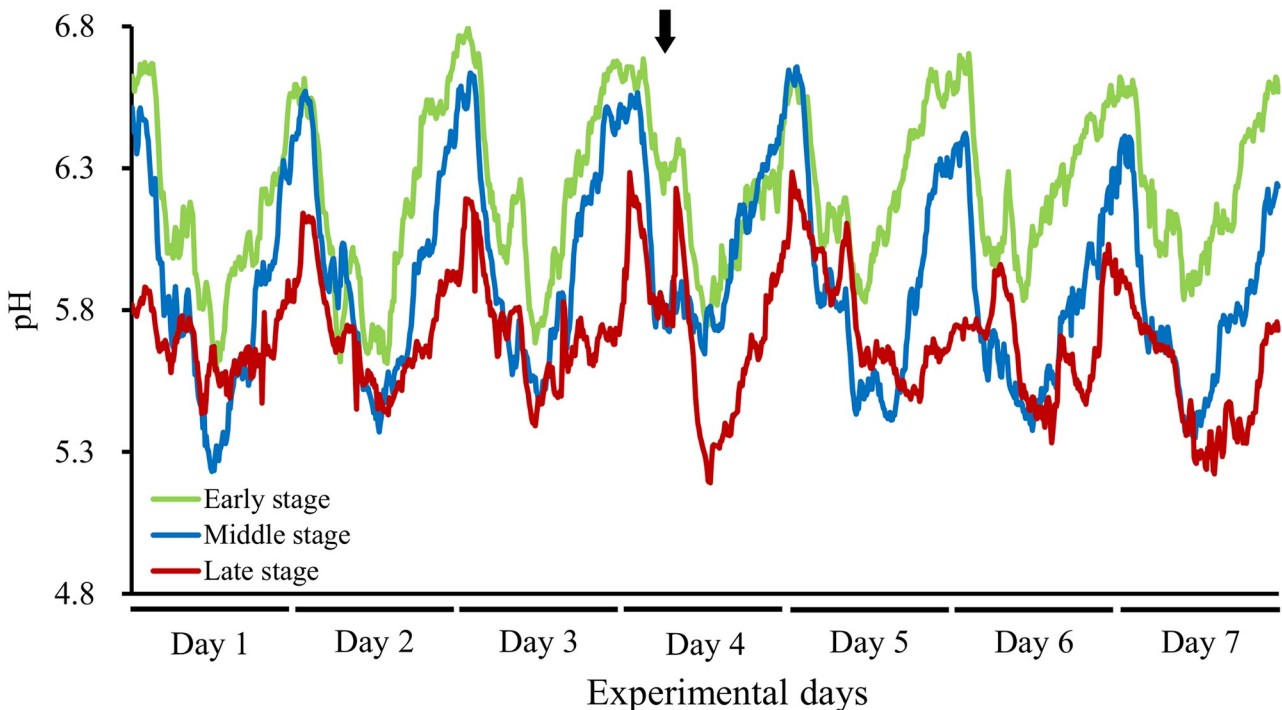

**Fig 1. Diurnal changes in the 10-minute mean ruminal pH in Japanese Black beef cattle.** Days 1–7 correspond to observations made during the final 7 days of each fattening stage. Arrows indicate the sample collection time (1300 h).

**Table 3. Total VFA, individual VFA proportions, acetic acid to propionic acid (A/P) ratio, lactic acid concentrations, and LPS activity in Japanese Black beef cattle during the Early, Middle, and Late fattening stages.**

| Item | Stage | | | SEM |
|---|---|---|---|---|
| | Early | Middle | Late | |
| Total VFA (mmol/dL) | 13.1[a] | 12.3[a] | 9.77[b] | 0.72 |
| Acetic acid (%) | 62.5 | 57.1 | 58.6 | 1.69 |
| Propionic acid (%) | 21.4 | 27.1 | 27.1 | 2.08 |
| Butyric acid (%) | 11.9 | 12.8 | 11.2 | 0.92 |
| Other (%) | 4.20 | 3.04 | 3.08 | 0.53 |
| A/P ratio | 3.06[a] | 2.24[b] | 2.34[ab] | 0.24 |
| Lactic acid (mmol/dL) | 0.75[a] | 0.28[b] | 1.57[c] | 0.10 |
| LPS (×10⁴ EU/mL) | 1.34[a] | 4.29[b] | 6.62[b] | 1.49 |

[a,b,c]Mean within a row, different superscripts significantly differ ($P < 0.05$)

during the Early stage was significantly higher than during the Middle stage ($P < 0.05$). The lactic acid concentration during the Late stage was significantly higher than during the Early and Middle stages ($P < 0.05$), and that during the Middle stage was significantly lower than during the Early stage ($P < 0.05$). Ruminal LPS activity during the Early stage was significantly lower than during the Middle and Late stages ($P < 0.05$; Table 3).

Serum AST activity during the Late stage was significantly higher than during the Early and Middle stages ($P < 0.05$; Table 4). During the Late fattening stage, the vitamin A level was significantly higher ($P < 0.05$), and β-carotene and vitamin E levels were significantly lower ($P < 0.05$) than during the Early and Middle stages.

## Principal component analyses of rumen parameters and peripheral blood metabolites

The 24-h mean and minimum ruminal pH were the dominant factors influencing ruminal pH parameters in the Early stage, and the duration of time where pH < 5.6 and 5.8, and AUC

**Table 4. Peripheral blood metabolite analysis in Japanese Black beef cattle during the Early, Middle, and Late fattening stages.**

| Item[1] | Stage | | | SEM |
|---|---|---|---|---|
| | Early | Middle | Late | |
| TP (g/dL) | 6.68 | 6.89 | 6.96 | 0.18 |
| BUN (mg/dL) | 12.1 | 14.5 | 11.3 | 0.98 |
| TCHO (mg/dL) | 106 | 109 | 93.0 | 7.31 |
| AST (IU/L) | 69.2[a] | 71.3[a] | 100[b] | 10.1 |
| GGT (IU/L) | 27.7 | 26.9 | 25.5 | 3.18 |
| Ca (mg/dL) | 9.97 | 9.84 | 9.72 | 0.12 |
| Vitamin A (IU/dL) | 40.2[a] | 35.5[a] | 62.4[b] | 3.67 |
| β-carotene (μg/dL) | 0.48[a] | 0.33[a] | 0.11[b] | 0.06 |
| Vitamin E (μg/dL) | 122[a] | 171[a] | 92.6[b] | 12.5 |
| LBP (ng/mL) | 283 | 493 | 389 | 67.3 |

[a,b]Mean within a row, different superscripts differ ($P < 0.05$)

[1]TP = total protein; BUN = blood urea nitrogen; TCHO = total cholesterol; AST = aspartate transaminase; GGT = γ-glutamyl transpeptidase; Ca = calcium; LBP = lipopolysaccharide binding protein.

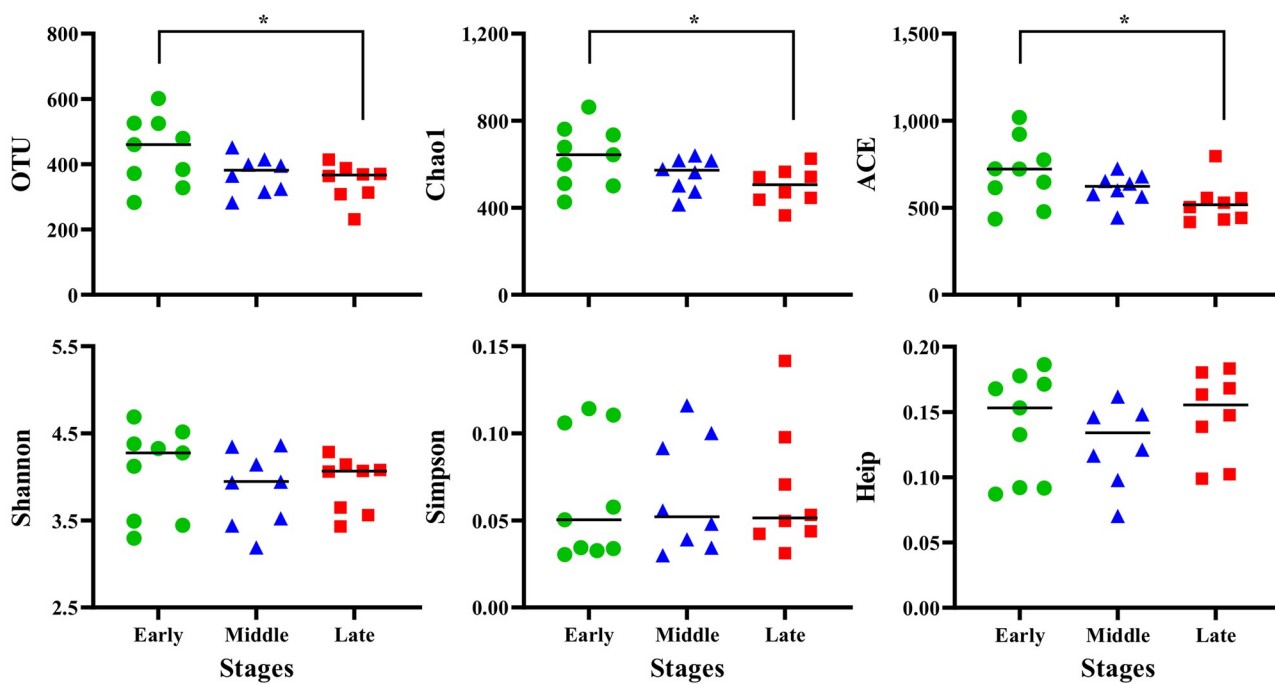

**Fig 2. Principal component analysis (PCA) plots for Japanese Black beef cattle.** PCA plots were generated for ruminal pH parameters (A), rumen fermentation parameters (B), and peripheral blood metabolites (C). PC1 and PC2 represent principal components 1 and 2, respectively.

values for pH < 5.6 and 5.8, were the most influential variables during the Late stage (principal components 1 + 2, explaining 82.2% of the variance; Fig 2A). The proportions of acetic and butyric acids were the most dominant factors influencing rumen fermentation parameters in the Early stage, and lactic acid concentration and ruminal LPS activity were the most influential factors during the Late stage (principal components 1 + 2, explaining 57.7% of the variance; Fig 2B). In the peripheral blood metabolites, PCA plots showed that the Early and Middle stage were most influenced by T-CHO, vitamin E, β-carotene, BUN, GGT, and Ca, and the Late stage was most affected by vitamin A and AST (principal components 1 + 2, explaining 50.7% of the variance; Fig 2C).

## Bacterial richness and diversity analysis

Bacterial richness indices (OTU, Chao1, and ACE) showed a gradual decrease from the Early to Late stages, and bacterial richness during the Late stage was significantly lower than during the Early stage ($P < 0.05$; Fig 3). However, bacterial diversity indices (Shannon, Simpson, and Heip) did not differ among the Early, Middle, and Late stages.

## Non-metric multidimensional scaling analyses of OTUs and KEGG pathway categories

The NMDS plots of OTUs during the Early stage showed a relatively clustered appearance compared with those during the Middle and Late stages, while those of OTUs during the Middle stage showed the most dispersed distribution (Fig 4A). Similarly, the NMDS plots of KEGG pathway categories during the Middle stage also showed a more scattered distribution compared with the Early and Late stage plots (Fig 4B). The stress of NMDS analysis was 0.10 for the OTU-based ordination and 0.14 for the KEGG pathway-based ordination.

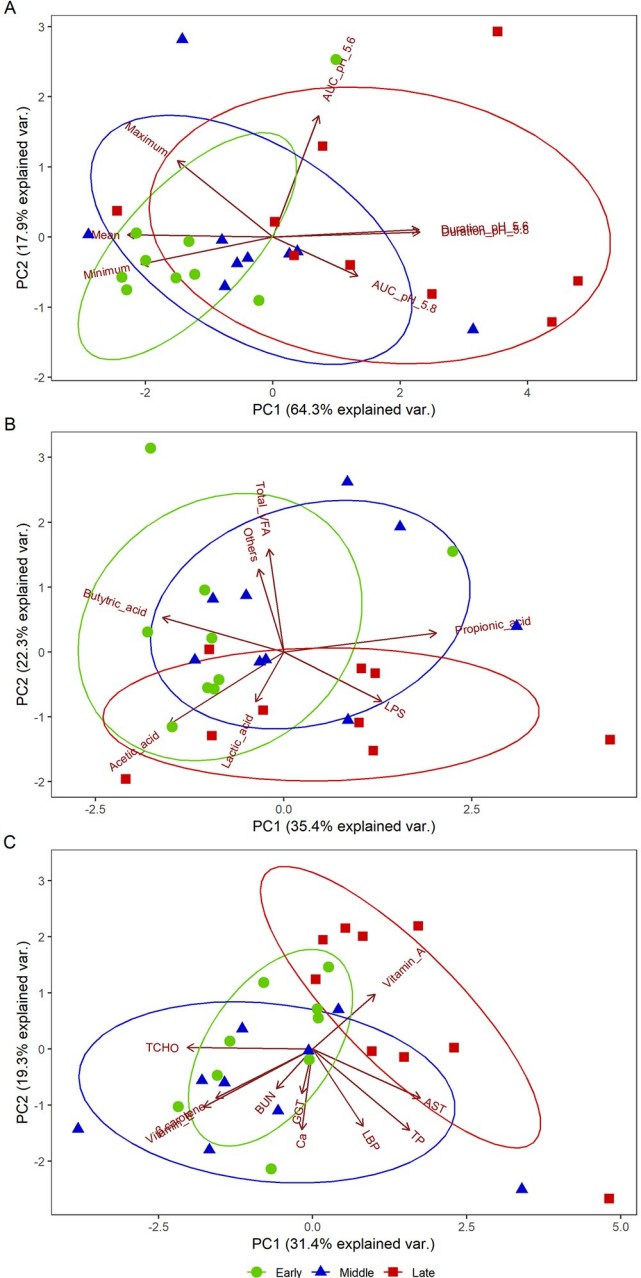

**Fig 3. Column scatter plots of bacterial richness and diversity indices.** The Mothur program (version 1.41.1; University of Michigan; http://www.mothur.org/wiki/; Schloss et al., 2009) was used to analyze the bacterial richness (operational taxonomic unit; OTU, Chao1, and abundance-based coverage estimator; ACE) and diversity (Shannon, Simpson, and Heip) indices. *significant difference at $P < 0.05$.

## Relative abundances of core bacterial OTUs

OTU1 (unclassified *Ruminococcaceae*) and OTU2 (unclassified *Lachnospiraceae*) were the most abundant (% of total sequence data) OTUs in the bacterial community. The relative abundances of OTU5 (unclassified *Firmicutes*), OTU8 (unclassified *Ruminococcaceae*), OTU26 (*Butyrivibrio*), OTU30 (unclassified *Firmicutes*), OTU37 (Unclassified *Clostridiales Incertae Sedis XIII*), OTU110 (Unclassified *Clostridiales*), OTU189 (Unclassified *Firmicutes*),

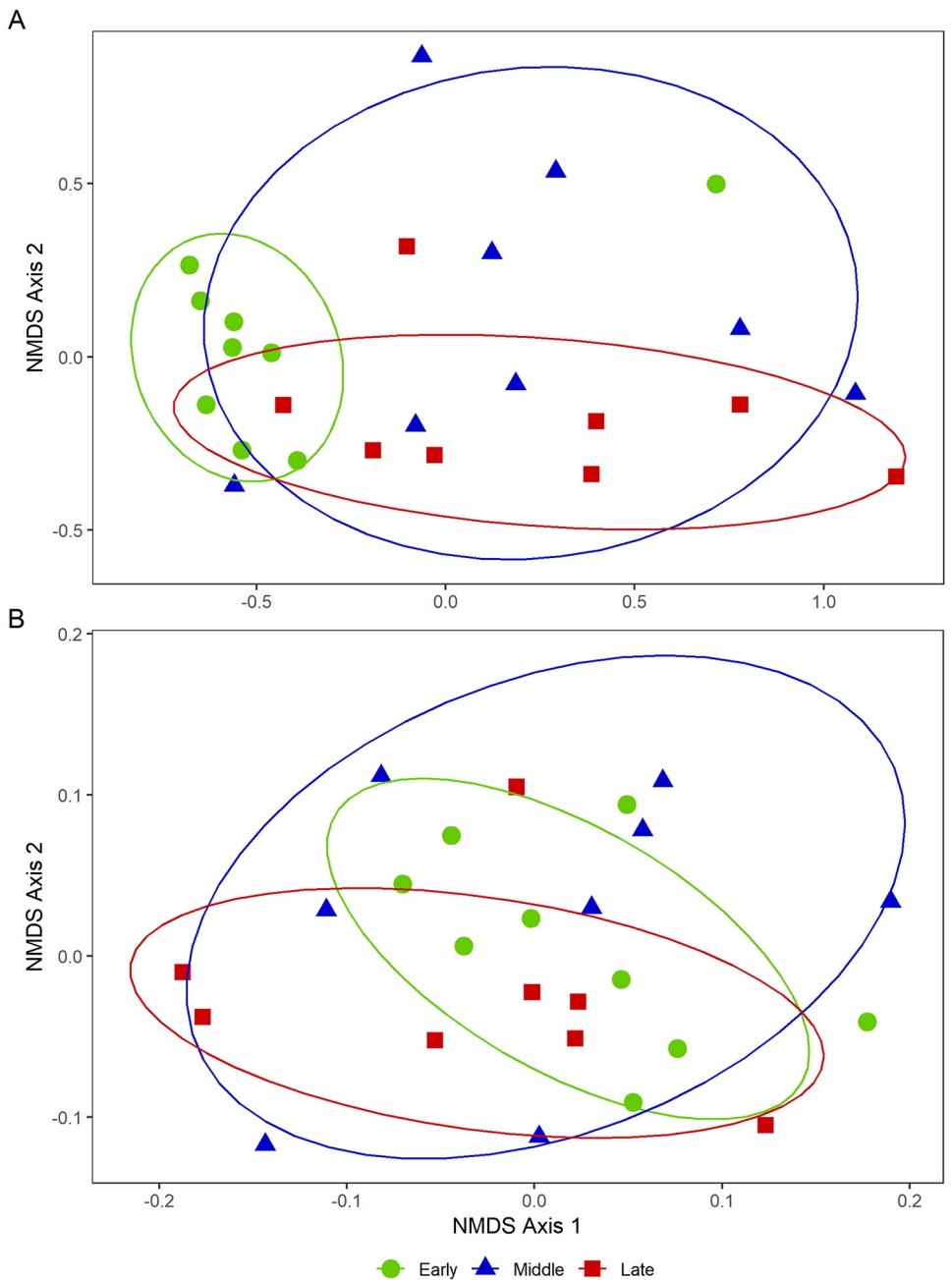

A

B

● Early  ▲ Middle  ■ Late

**Fig 4. Non-metric multidimensional scaling (NMDS) plots for Japanese Black beef cattle.** NMDS plots were generated for the bacterial OTUs (A) and Kyoto Encyclopedia of Genes and Genomes (KEGG) pathway categories (B). The stress of NMDS analysis was 0.10 and 0.14 for the OTU- and KEGG pathway-based ordinations, respectively.

and OTU238 (Unclassified *Lachnospiraceae*) during the Early stage were significantly higher ($P < 0.05$), and those of OTU43 (unclassified *Lachnospiraceae*), OTU55 (Unclassified *Firmicutes*), and OTU62 (*Ruminococcus*) during the Early stage were significantly lower ($P < 0.05$), than those during the Middle stage (Table 5). The relative abundances of OTU26, OTU30, OTU37, OTU43, and OTU199 (unclassified *Lachnospiraceae*) during the Early stage were significantly higher ($P < 0.05$), and those of OTU55 and OTU167 (*Intestinimonas*) during the

**Table 5. Relative abundances and taxonomic classification of core operational taxonomic units (OTU; shared by all samples) in Japanese Black beef cattle during the Early, Middle, and Late fattening stages.**

| OTU | Stage | | | SEM | RDP[1] classification (genus level) | BLASTn[2] classification | Percent to BLASTn identity[3] | Accession no. |
|---|---|---|---|---|---|---|---|---|
| | Early | Middle | Late | | | | | |
| OTU1 | 11.7 | 6.73 | 6.84 | 3.05 | Unclassified *Ruminococcaceae* | *Ruminococcus bromii* strain ATCC 27255 | 97.8 | NR_025930.1 |
| OTU2 | 6.85 | 6.47 | 6.46 | 2.31 | Unclassified *Lachnospiraceae* | *Faecalimonas umbilicata* strain EGH7 | 95.2 | NR_156907.1 |
| OTU4 | 2.89 | 3.46 | 2.86 | 1.42 | *Succiniclasticum* | *Succiniclasticum ruminis* strain SE10 | 95.9 | NR_026205.1 |
| OTU5 | 5.14[a] | 1.85[b] | 2.14[ab] | 0.77 | Unclassified *Firmicutes* | *Thermotalea metallivorans* strain B2-1 | 88.6 | NR_044503.1 |
| OTU6 | 2.31 | 3.16 | 1.40 | 0.84 | *Prevotella* | *Prevotella ruminicola* strain Bryant 23 | 98.9 | NR_102887.1 |
| OTU8 | 4.62[a] | 1.37[b] | 1.91[b] | 0.45 | Unclassified *Ruminococcaceae* | *Pseudoflavonifractor phocaeensis* strain Marseille-P3064 | 92.2 | NR_147370.1 |
| OTU10 | 2.10 | 2.04 | 0.80 | 0.57 | Unclassified *Ruminococcaceae* | *Ruminococcus bromii* strain ATCC 27255 | 95.6 | NR_025930.1 |
| OTU13 | 1.66 | 2.01 | 1.48 | 0.42 | *Mogibacterium* | *Mogibacterium neglectum* strain P9a-h | 94.8 | NR_027203.1 |
| OTU15 | 1.77 | 0.69 | 2.26 | 0.66 | Unclassified *Ruminococcaceae* | *Ruminococcus bromii* strain ATCC 27255 | 94.5 | NR_025930.1 |
| OTU24 | 0.98 | 0.53 | 0.72 | 0.27 | Unclassified *Lachnospiraceae* | *Faecalicatena orotica* strain JCM 1429 | 93.3 | NR_114392.1 |
| OTU26 | 0.97[a] | 0.17[b] | 0.20[b] | 0.14 | *Butyrivibrio* | *Butyrivibrio proteoclasticus* strain B316 | 99.3 | NR_102893.1 |
| OTU30 | 1.13[a] | 0.13[b] | 0.35[c] | 0.11 | Unclassified *Firmicutes* | *Novibacillus thermophilus* strain SG-1 | 87.4 | NR_136797.1 |
| OTU34 | 0.65 | 0.28 | 0.71 | 0.17 | Unclassified *Firmicutes* | *Monoglobus pectinilyticus* strain 14 | 90.0 | NR_159227.1 |
| OTU35 | 0.56 | 0.62 | 0.77 | 0.29 | *Olsenella* | *Olsenella profusa* DSM 13989 | 98.5 | NR_116938.1 |
| OTU37 | 0.70[a] | 0.14[b] | 0.34[b] | 0.10 | Unclassified *Clostridiales Incertae Sedis XIII* | *Emergencia timonensis* strain SN18 | 93.3 | NR_144737.1 |
| OTU43 | 0.38[a] | 0.48[b] | 0.20[b] | 0.13 | Unclassified *Lachnospiraceae* | *Acetatifactor muris* strain CT-m2 | 91.5 | NR_117905.1 |
| OTU54 | 0.26 | 0.38 | 0.19 | 0.10 | *Schwartzia* | *Schwartzia succinivorans* strain S1-1 | 99.3 | NR_029325.1 |
| OTU55 | 0.12[a] | 0.59[b] | 0.57[b] | 0.17 | Unclassified *Firmicutes* | *Salinithrix halophila* strain R4S8 | 87.0 | NR_134171.1 |
| OTU58 | 0.20 | 0.46 | 0.21 | 0.12 | Unclassified *Firmicutes* | *Gracilibacter thermotolerans* JW/YJL-S1 | 89.6 | NR_115693.1 |
| OTU62 | 0.09[a] | 0.45[b] | 0.12[a] | 0.06 | *Ruminococcus* | *Ruminococcus flavefaciens* strain C94 | 98.9 | NR_025931.1 |
| OTU64 | 0.19 | 0.41 | 0.36 | 0.15 | Unclassified *Planctomycetaceae* | *Thermostilla marina* strain SVX8 | 84.8 | NR_148598.1 |
| OTU68 | 0.23 | 0.24 | 0.89 | 0.19 | Unclassified *Lachnospiraceae* | *Blautia glucerasea* strain JCM 17039 | 93.0 | NR_113231.1 |
| OTU79 | 0.21 | 0.27 | 0.30 | 0.09 | Unclassified *Clostridiales* | *Ihubacter massiliensis* strain Marseille-P2843 | 93.7 | NR_144749.1 |
| OTU80 | 0.24 | 0.28 | 0.12 | 0.07 | Unclassified *Firmicutes* | *Geosporobacter ferrireducens* strain IRF9 | 88.6 | NR_148302.1 |
| OTU86 | 0.14 | 0.21 | 0.23 | 0.08 | Unclassified *Clostridiales* | *Vallitalea pronyensis* strain FatNI3 | 90.4 | NR_125677.1 |
| OTU90 | 0.27 | 0.13 | 0.24 | 0.07 | *Atopobium* | *Atopobium parvulum* DSM 20469 | 95.5 | NR_102936.1 |
| OTU103 | 0.14 | 0.12 | 0.22 | 0.07 | Unclassified *Ruminococcaceae* | *Sporobacter termitidis* strain SYR | 93.0 | NR_044972.1 |
| OTU110 | 0.23[a] | 0.04[b] | 0.10[ab] | 0.05 | Unclassified *Clostridiales* | *Anaerobacterium chartisolvens* strain T-1-35 | 89.3 | NR_125464.1 |
| OTU125 | 0.09 | 0.12 | 0.23 | 0.05 | Unclassified *Ruminococcaceae* | *Ruminococcus flavefaciens* strain C94 | 94.1 | NR_025931.1 |
| OTU167 | 0.08[a] | 0.06[a] | 0.17[b] | 0.02 | *Intestinimonas* | *Intestinimonas butyriciproducens* strain SRB-521-5-I | 96.7 | NR_118554.1 |
| OTU183 | 0.09 | 0.04 | 0.07 | 0.02 | Unclassified *Clostridiales* | *Eubacterium nodatum* ATCC 33099 | 91.8 | NR_118781.1 |
| OTU184 | 0.03 | 0.09 | 0.08 | 0.02 | *Ruminococcus* | *Ruminococcus albus* 7 = DSM 20455 | 98.5 | NR_074399.1 |
| OTU189 | 0.09[a] | 0.04[b] | 0.09[ab] | 0.02 | Unclassified *Firmicutes* | *Caloramator fervidus* strain RT4. B1 | 89.6 | NR_025899.1 |
| OTU199 | 0.07[a] | 0.06[a] | 0.02[b] | 0.01 | Unclassified *Lachnospiraceae* | [Clostridium] *aminophilum* strain F | 93.3 | NR_118651.1 |
| OTU238 | 0.07[a] | 0.02[b] | 0.08[a] | 0.02 | Unclassified *Lachnospiraceae* | *Merdimonas faecis* strain BR31 | 91.1 | NR_157642.1 |

[a,b,c]Mean within a row, different superscripts significantly differ ($P < 0.05$)

[1]Ribosomal Database Project tools training set version 16 in the MiSeq standard operating procedure (MiSeq SOP) in Mothur (https://mothur.org/wiki/MiSeq_SOP; [19])

[2]Basic Local Alignment Search Tool

[3]Percent identity

Early stage were significantly lower ($P < 0.05$), than during the Late stage. The relative abundances of OTU30, OTU167, and OTU238 during the Middle stage were significantly lower ($P < 0.05$), and those of OTU62 and OTU199 during the Middle stage was significantly higher ($P < 0.05$), than during the Late stage.

### Pearson correlation analyses of rumen parameters and core bacterial OTUs

Among the OTUs that were significantly correlated with rumen fermentation parameters ($P < 0.05$), the relative abundances of OTU1 (unclassified *Ruminococcaceae*), OTU8 (unclassified *Ruminococcaceae*), and OTU26 (*Butyrivibrio*) were positively correlated with the 24-h mean ruminal pH (r = 0.416, 0.427, and 0.476, respectively) and 24-h minimum ruminal pH (r = 0.458, 0.454, and 0.435, respectively), while they were negatively correlated with the duration of time where pH < 5.6 (r = -0.476, -0.472, and -0.432, respectively) and < 5.8 (r = -0.509, -0.476, and -0.473, respectively) ([Fig 5]). In contrast, the relative abundance of OTU68 (unclassified *Lachnospiraceae*) was negatively correlated with the 24-h mean (r = -0.530), minimum (r = -0.531), and maximum (r = -0.499) ruminal pH, and positively correlated with the duration of time where pH < 5.6 (r = 0.453) and < 5.8 (r = 0.417). Total VFA concentration was negatively correlated with OTU167 (r = -0.412, $P < 0.05$). Lactic acid concentration was positively correlated with OTU34 (unclassified *Firmicutes*; r = 0.521), OTU68 (r = 0.503), OTU167 (r = 0.556), and OTU238 (r = 0.587) and negatively correlated with OTU6 (r = -0.449; *Prevotella*) and OTU62 (r = -0.453; all $P < 0.05$). The ruminal LPS activity was positively correlated with OTU125 (r = 0.519; unclassified *Ruminococcaceae*) and negatively correlated with OTU5 (r = -0.413), OTU8 (r = -0.440), OTU13 (r = -0.414; *Mogibacterium*), and OTU37 (r = -0.470; all $P < 0.05$).

## Discussion

Japanese Black beef cattle are raised on a high-grain diet for about 20 months, i.e., between 10 and 30 months of age, to increase intramuscular fat accumulation (marbling score), and the fattening period consists of three fattening stages. We collected samples over the entire fattening period and explored long-term changes in ruminal pH and fermentation, as well as their consequences with respect to the rumen bacterial community.

The occurrence of SARA may cause various health problems in cattle, such as feed intake depression, reduced fiber digestion, milk fat depression, diarrhea, laminitis, liver abscesses, increased production of bacterial endotoxins, and inflammation [25]. In the present study, however, the cattle showed no clinical sign of abnormal body condition, such as high body temperature, acute feed intake, dehydration, and diarrhea, throughout the study period, and the body weight of them showed a gradual but significant increase across the three fattening stages. Dietary intake amounts were highest during the Middle stage and lowest during the Late stage as an adaptation to long-term high-grain diet feeding or response to significantly lowered ruminal pH during the latter fattening stage. However, growth performance during the Late stage was not impaired, and changes in the 24-h mean ruminal pH were not consistent with dietary intake or rates of DM and TDN. The 24-h mean ruminal pH gradually decreased toward the end of the fattening period. Although the 24-h mean pH value during the Early and Middle stages (6.22 and 6.06, respectively) were similar level compared with the SARA challenge model for 2 days (5.94 and 5.81; [26]), for 1 week (6.10; [3]) and mid-term for 6 weeks (5.97; [4]) in Holstein cattle studies, Japanese Black cattle presented more severe depression of ruminal pH (5.73) during the Late stage in the present study. Furthermore, the total VFA concentration, where VFAs are the most abundant organic acids in the rumen [3], is correlated negatively with ruminal pH (r = -0681, $P > 0.05$; [27]). However, total VFA concentration did

# Rumen Parameters

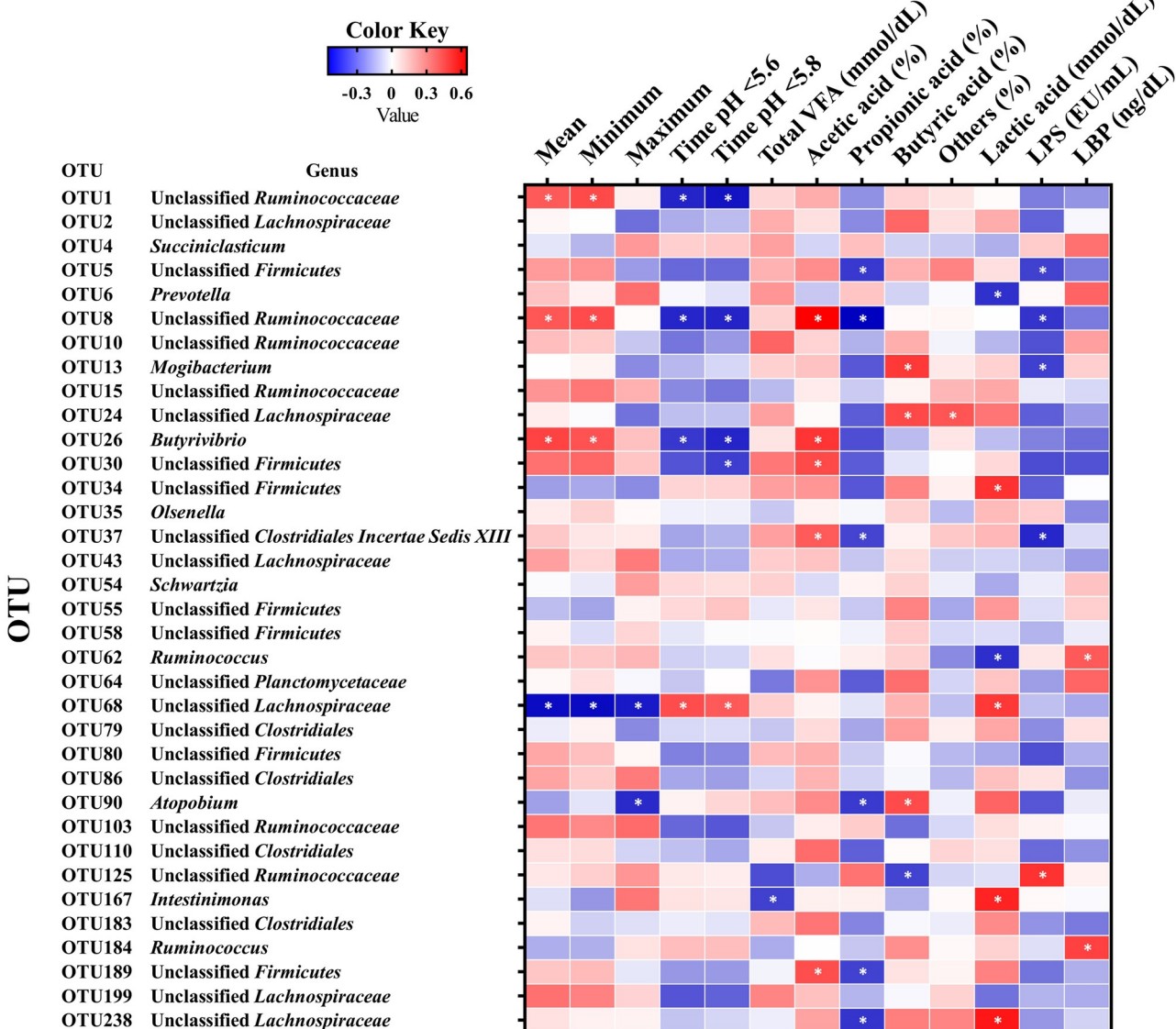

**Fig 5. Correlation analyses between the core OTUs (shared by all samples) and rumen parameters.** Cells are colored based on Pearson correlation analyses. Blue represents a negative correlation and red represents a positive correlation. *significant correlation between OTUs and rumen parameters at $P < 0.05$. Mean = 24-h mean ruminal pH; Minimum = 24-h minimum ruminal pH; Maximum = 24-h maximum ruminal pH; Time pH < 5.6 = duration of time where pH < 5.6; Time pH < 5.8 = duration of time where pH < 5.8; LPS = lipopolysaccharide; LBP = lipopolysaccharide-binding protein.

not accord with the changes in ruminal pH parameters (r = 0.258, $P > 0.05$), and gradually decreased toward the end of the fattening period. The significantly higher lactic acid concentration during the Late stage may play a role in the lowered ruminal pH during the same stage, suggesting that different mechanisms underlie low ruminal pH values occurring in the Late versus Middle and Early stages due to 10 times less protonated lactic acid property than VFA (pKa 4.9 *vs.* 3.9) [3]. Furthermore, decrease and increase in the proportions of acetic and propionic acids, respectively, were consistent with general feature of high-grain diet feeding in Holstein cattle [3, 4]. The PCA plots showed that lactic acid and LPS were the most influential

variables during the Late stage (Fig 3B). The Japanese Black beef cattle suffered from SARA due to higher lactic acid levels during the Late stage, suggestive of a partial transition from VFA production to lactic acid production in the rumen or enhanced absorption of VFA by rumen epithelium transporters (i.e. sodium hydrogen exchanger isoform 3; [28], and mono-carboxylate transporter isoform 4; [29]). However, lower ruminal pH during the Late stage did not disrupt the gastrointestinal barrier or induce higher LBP levels in the peripheral blood, despite inducing significantly higher LPS levels in the rumen [30, 31].

In the present study, the cattle were typically healthy, with no recent antibiotic use during fistulation surgery (performed at 12 months of age) and no signs of obvious illness during the experimental period; thus, there was no indication of negative effects of antibiotics or fistulation surgery on the rumen bacterial community. Regarding the relationships of ruminal pH, bacterial diversity and richness indices, correlations of pH parameters with bacterial diversity and richness were generally positive; low ruminal pH leads to low bacterial diversity and richness [3, 4, 6, 7]. In the present study, the decrease in ruminal pH was consistent with the decrease in bacterial richness (OTU, Chao1, and ACE), but not with the Shannon, Simpson, and Heip bacterial diversity indices. The mean bacterial diversity and richness indices during the Early stage were similar to those in a short-term SARA challenge model of Holstein cattle with a similar sampling size (4,623 sequences/sample; [3]). This suggests that ruminal pH during the Early stage was low (SARA "challenge level") and gradually decreased during the latter fattening period, thus reducing ruminal bacterial richness but not bacterial diversity. To best our knowledge, this is the first study demonstrating the relationship between the long-term high-grain diet feeding and bacterial diversity or richness, and suggests that long-term high-grain diet consumption results in the preservation of bacterial diversity to protect against dysbiosis of the entire rumen bacterial community in Japanese Black beef cattle.

NMDS plots showed the taxonomic and genetic structures of the rumen bacterial communities based on OTUs and KEGG pathway categories. Previously, short-term high-grain diet feeding of Holstein cattle was associated with dispersed principal coordinate analysis plot data, with lower ruminal pH and higher VFA concentrations compared with control diet [26, 32, 33]. In the present study, NMDS plots for the Middle fattening stage showed a scattered appearance for both OTU and KEGG data. Therefore, we suggest that the bacterial community during the Middle stage was in the process of adapting to long-term high-grain diet feeding, before producing more lactic acid in the rumen of Japanese Black beef cattle during the Late fattening stage. Further studies are required to fully explore the relationships of KEGG categories (predicted functional pathway) and ruminal pH or fermentation parameters.

The present study showed that a total of 35 OTUs (core microbiota) were shared by all samples and all fattening stages. In addition, taxonomic classification was performed against the RDP training set; further classification was performed using the GenBank database, to assign OTUs to a specific taxonomic level. In the present study, unclassified *Ruminococcaceae*, unclassified *Lachnospiraceae*, and *Prevotella* were the most abundant genera in Japanese Black cattle, which is not consistent with previous studies showing that the genus *Prevotella* was generally the most predominant in the rumen bacterial community of Holstein cattle [3, 7, 16, 34]. This is because different breeds of cattle may have different feed passage rate through the digestive tract due to different eating and rumination behaviors [35], and Holstein cows are fed high-grain based diet to maximize productivity, but considering health and reproductivity [36], compared with those fed to maximize productivity and to produce beef with quality [11]. Furthermore, the relative abundances of several OTUs showed significant changes across the fattening stages. For example, changes in the proportion of OTUs were consistent with changes in the ruminal pH and fermentation parameters based on Pearson correlation analysis; the relative abundances of OTU8 (Family *Ruminococcaceae*) and OTU26 (Genus *Butyrivibrio*) were

positively correlated with the 24-h mean ruminal pH. The unclassified *Ruminococcaceae* and unclassified *Lachnospiraceae* have been associated with the maintenance of gut health and play a role as active plant degraders [37], where the *Ruminococcus* may contribute to starch fermentation [3] and *Butytivibrio* is dominant in the rumen due to fermentation of a range of substrates [38]. In addition, both OTU167 and OTU238 proportions were positively correlated with the lactic acid concentration. OTU238 (Family *Lachnospiraceae*; *Merdimonas faecis strain BR31*) may be associated with the production of lactic acid in the rumen [39], and increased lactic acid concentrations may be exploited by OTU167 (Genus *Intestinimonas*; *Intestinimonas butyriciproducens strain SRB-521-5-I*) as its primary energy and carbon source [40]. Collectively, changes in bacterial richness, structure, and composition during the fattening period may represent an adaptation to long-term high-grain diet feeding, and was shown to affect or be affected by changes in ruminal pH and fermentation. Furthermore, the core microbiota in the present study may have arisen as a consequence of adaptation to and endurance of a long-term, harsh ruminal environment, to protect against dysbiosis of the entire rumen bacterial community, resulting in unique and distinct microbiota in Japanese Black beef cattle.

## Conclusions

Long-term high-grain feeding (from 10 to 30 months of age) in Japanese Black cattle induced a gradual decrease in ruminal pH and total VFA production during the latter fattening period. In contrast, the lactic acid concentration during the Late stage increased significantly compared with the earlier stages, suggestive of a different underlying mechanism of SARA. Regarding the rumen bacterial composition, the unclassified *Ruminococcaceae* and unclassified *Lachnospiraceae* were the most abundant bacterial genera, while family *Lachnospiraceae* (OTU238) and genus *Intestinimonas* (OTU167) may be associated with lactic acid production and utilization, respectively, during the Late stage. Taken together, the specialized fattening technique applied herein to Japanese Black beef cattle resulted in unique changes in the rumen fermentation characteristics and bacterial community composition, as adaptations to long-term high-grain diet feeding.

## Author Contributions

**Conceptualization:** Toru Ogata, Shigeru Sato.

**Data curation:** Eiji Iwamoto, Tatsunori Masaki, Kentaro Ikuta, Yo-Han Kim.

**Formal analysis:** Hiroki Makino, Naoki Ishizuka, Eiji Iwamoto, Yo-Han Kim.

**Investigation:** Hiroki Makino, Naoki Ishizuka, Eiji Iwamoto, Tatsunori Masaki, Kentaro Ikuta.

**Methodology:** Tatsunori Masaki, Kentaro Ikuta, Yo-Han Kim.

**Project administration:** Toru Ogata, Shigeru Sato.

**Resources:** Eiji Iwamoto, Tatsunori Masaki, Kentaro Ikuta.

**Supervision:** Shigeru Sato.

**Validation:** Yo-Han Kim.

**Visualization:** Yo-Han Kim.

**Writing – original draft:** Toru Ogata, Yo-Han Kim, Shigeru Sato.

**Writing – review & editing:** Toru Ogata, Yo-Han Kim, Shigeru Sato.

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
