## [Decision Letter · Decision Letter 0]

3 Sep 2019

PONE-D-19-20947

Long-term high-grain diet altered the ruminal pH, fermentation, and composition and functions of the rumen bacterial community, leading to enhanced lactic acid production in Japanese Black beef cattle during fattening

PLOS ONE

Dear Dr Shigeru Sato,

Thank you for submitting your manuscript to PLOS ONE. After careful consideration, we feel that it has merit but does not fully meet PLOS ONE’s publication criteria as it currently stands. Therefore, we invite you to submit a revised version of the manuscript that addresses the points raised during the review process.

We would appreciate receiving your revised manuscript by October 4. To enhance the reproducibility of your results, we recommend that if applicable you deposit your laboratory protocols in protocols.io, where a protocol can be assigned its own identifier (DOI) such that it can be cited independently in the future. For instructions see: http://journals.plos.org/plosone/s/submission-guidelines#loc-laboratory-protocols

We look forward to receiving your revised manuscript.

Kind regards,

Marcio de Souza Duarte

Academic Editor

PLOS ONE

Journal Requirements:

2. At this time, we request that you  please report additional details in your Methods section regarding animal care: 1) Please provide details of animal welfare (e.g., shelter, food, water, environmental enrichment), including the name and location where the animals were housed. 2) please describe any steps taken to minimize animal suffering and distress, such as by administering analgesics or anesthetics, and 3) Please explain whether the animals were euthanized at the completion of the study and, if so, include the method of sacrifice. Thank you for your attention to these requests.

Reviewers' comments:

Reviewer's Responses to Questions

**Comments to the Author**

1. Is the manuscript technically sound, and do the data support the conclusions?

Reviewer #1: Partly

Reviewer #2: Yes

2. Has the statistical analysis been performed appropriately and rigorously? 

Reviewer #1: Yes

Reviewer #2: Yes

3. Have the authors made all data underlying the findings in their manuscript fully available?

Reviewer #1: Yes

Reviewer #2: Yes

4. Is the manuscript presented in an intelligible fashion and written in standard English?

Reviewer #1: Yes

Reviewer #2: Yes

5. Review Comments to the Author

Reviewer #1: General comments

The paper intitled ‘Long-term high-grain diet altered the ruminal pH, fermentation, and composition and functions of the rumen bacterial community, leading to enhanced lactic acid production in Japanese Black beef cattle during fattening’ described rumen characteristics, such as pH, VFA, and lactic acid, blood metabolites, and rumen microbial abundance and diversity of Japanese Black beef cattle during Early, Mid and Late fattening stages.

The manuscript is well-prepared

The novelty about the work lies on the evaluation of rumen characteristics of cattle which are fed high levels of concentrate from 10 to 30 months, and how animals can cope the diet.

The study has one major flaw which should be addressed. According to experimental design described, animals were on considered on Early fattening stage from 10 to 14 months. However, animals were fistulated when they were 12 months old. Thus, considering at least on month for recovery in the best case scenario, and another 3 weeks for diet adaption, different fattening stages should be considered.

Also, minor considerations should be the forage-to-concentrate ratio that was modified throughout the experiment, being 26:74 during Early stage, 13:87 during the Middle stage, and 14:86 during Late stage, which might have affected the rumen environment and pH values. Also, the diet composition should be presented, and the methods used for chemical analysis. Vitamin A concentration should also be included in the Table 1, as you discussed hypovitaminosis in the Introduction section.

Finally, the lack of mechanical lysis of rumen content is troublesome and data interpretation and extrapolation should be made carefully. Several studies have shown that disruption of bacteria with tough cell walls is more efficient with a mechanical approach than by an enzyme-based protocol. Furthermore, extraction methods have important implications on the results, and studies using different extraction procedures should not be compared. For that, the conclusion needs to be re-worked.

Despite these considerations, the paper is well-prepared. Studies exploring the rumen microbiome and metabolic disorders are needed.

If necessary, I would be available to look at the revised version.

Specific comments

Please, verify financial disclosure guidelines and amend if appropriate. According to PLOS ONE guidelines, Funded studies should have statements with the following details: Initials of the authors who received each award; Grant numbers awarded to each author; The full name of each funder; URL of each funder website; and whether the sponsors or funders play any role in the study design, data collection and analysis, decision to publish, or preparation of the manuscript.

P02L26: what ‘specialized high-concentrate diets’ means?

P02L29: the term ‘were collected on day 4 of pH measurement’ is confusing.

P02L45-46: I respectfully disagree with the first sentence of the Introduction section, as pasture based diet can also promote growth, productivity, and high-quality meat or milk.

P04L48-50: there are parts of the manuscript which are confusing, such as the phrase ‘As a result, ruminal pH decreases; subacute ruminal acidosis (SARA) and ruminal acidosis (RA) are defined by ruminal pHs of ≤ 5.6 and below, respectively (Nagaraja and Titgemeyer, 2007).’. For example, I would suggest modifying to ‘ruminal pHs values of ≤ 5.6’.

P05L69-71: this phrase should be re-worked as you did not evaluate the ‘effect of ruminal pH, bacterial community and fermentation on the fattening of 10-month-old Japanese Black beef cattle’. Rather, this work characterized ruminal pH, bacterial community and fermentation characteristics on different ages of Japanese Black beef cattle.

P05L71: ‘this findings’ is incorrect.

P06L82: what percentage of refusals?

P06L83: why 10–14, 15–22, and 23–30 months of age were selected as different stages? It seems arbitrary. This should be addressed.

P06L89: the rationale to feed concentrate after 1 hour of forage should be addressed. Also, how concentrate availability was ensured?

P06L93: the term ‘sufficient rate’ is dubious. Use other term.

P06L93: please, refer the Japanese feeding standard.

P08L104: the phrase ‘Rumen fluid samples were collected on day 4 of the pH measurements during the Early…’ is confusing and should be re-worked.

P09L122: there are two references of Kim et al. 2016, which should be differentiated according to PLOS ONE manuscript preparation guidelines.

P09L128: what was the ratio of phenol/chloroform/isoamyl alcohols?

P09L134: authors need to clarify whether pyrosequencing was the used approach. Furthermore, this should be addressed throughout the manuscript.

P11L175: the phrase ‘Rumen fluid samples were collected on day 4 of the pH measurements during the Early…’ is confusing and should be re-worked.

Table 2: please, use ‘ab’ to indicate similarities between stages. For example, the minimum pH values should be presented as ‘5.43a, 5.30ab, and 4.98b’ for Early, Middle, and Late stages, respectively. This should be considered for other tables.

Table 3: data on acetic, propionic and butyric acids should be discussed in the text.

P26L349: the effect of feed intake reduction in the Late stage should be discussed. Did animals decrease feed intake, and reduced concentrate intake by 20% to mitigate health problems?

P27L385: the discussion on reduction of ruminal bacterial richness but not bacterial diversity, should not be limited to one study. Please expand this discussion using other studies that corroborate with you study, other which does not, as it is central to your work. I would also consider bacterial and sequencing limitations, as bead beating was not used.

P28L401-405: the different core microbiota observed in your study compared to the literature was expected. A diet enriched in concentrate was used, having greater levels compared to dairy cattle diets. Furthermore, dairy cattle is anatomically different to beef cattle, which has implication of passage rate for example, and passage rate has huge influence on the rumen microbiota. Finally, generally beef cattle is fed to maximize productivity and to produce beef with quality. On the other hand, dairy cattle are fed to maximize productivity, but considering health and reproductivity. Thus, nutrient requirements and managements are different.

Reviewer #2: Line Comment

29 It this total VFA concentration?

72 Perhaps “understand” could be changed to “the understanding of”.

81, 374 It is noted that the cattle were rumen-fistulated at 12-mnths of age, during the Early stage of the trial. Please speculate whether this would have influenced the results obtained during that phase. It was also noted that fistualtion was said to be done at 10-months of age on line 374. If cattle were fistulated at 10 months, no discussion of affects on the animals is necessary.

85, 88, 90, etc. Suggest using either “roughage” or “forage” for that portion of the diet.

93, Table 1 What is “sufficient rate”? Is this a requirement? If it is a requirement, why are the units in “%”? Are the dairy intakes in kg of DM? Please clarify. Also, please supply a citation for the Japanese feeding standard.

117 Was the supernatant or the pellet analyzed for LPS activity? I ask because the LPS is presumably associated with the microbes and likely would be largely with the 11,000 x g pellet.

213-217 Again, please clarify what is meant by “rates”.

Table 2 Why are there no superscripts on several of the values for the Middle treatment (as is seen in Table 4)? Normally, one would expect that, if those values were not different from Early and/or Late, they would share the superscript with the Early and/or Late. Sometimes the Middle values are intermediate, or even greater than, either Early and Late.

Table 3 Why are there no superscripts for A/P ration for the Late treatment?

383-385 Do you mean to say that the risk of SARA during the Early stage was low, because pH was actually higher during this stage?

Additional comment Normally, one would expect to see a table of diet composition, showing the feedstuffs used to construct the 3 diets fed in this trial. The authors might consider adding a table containing this information to the paper.

6. PLOS authors have the option to publish the peer review history of their article (what does this mean?). If published, this will include your full peer review and any attached files.

Reviewer #1: No

Reviewer #2: No

---

## [Author Response · Author response to Decision Letter 0]

17 Sep 2019

PONE-D-19-20947

Long-term high-grain diet altered the ruminal pH, fermentation, and composition and functions of the rumen bacterial community, leading to enhanced lactic acid production in Japanese Black beef cattle during fattening

Dear Editor and Reviewers

 Authors would like to thank Editor and Reviewers for their helpful comments and suggestions. We have done our best to address all the issues raised by Reviewers very carefully in this first revision, which we believe has improved the quality of the paper further. To facilitate the reviewing process, we have highlighted all changes done by Authors (Yellow) in the revised manuscript. We have responded to every comment done by the Reviewers below, and also have indicated the changes made with respective new lines. 

 Before proceeding the present revision, we would like to declare that we have two corresponding authors who contributed equally to this work. The relevant changes are declared in the cover letter and title page of the manuscript. 

5. Review Comments to the Author

Reviewer #1: General comments

 The paper intitled ‘Long-term high-grain diet altered the ruminal pH, fermentation, and composition and functions of the rumen bacterial community, leading to enhanced lactic acid production in Japanese Black beef cattle during fattening’ described rumen characteristics, such as pH, VFA, and lactic acid, blood metabolites, and rumen microbial abundance and diversity of Japanese Black beef cattle during Early, Mid and Late fattening stages. The manuscript is well-prepared. The novelty about the work lies on the evaluation of rumen characteristics of cattle which are fed high levels of concentrate from 10 to 30 months, and how animals can cope the diet.

AU: Authors would like to thank Reviewer 1 for your helpful comments and suggestions. We have done our best to address all the issues raised by Reviewer 1 very carefully in this new revision, which we believe has improved the quality of the paper further. 

 The study has one major flaw which should be addressed. According to experimental design described, animals were on considered on Early fattening stage from 10 to 14 months. However, animals were fistulated when they were 12 months old. Thus, considering at least on month for recovery in the best case scenario, and another 3 weeks for diet adaption, different fattening stages should be considered.

AU: We would like to thank you for your comment. As you mentioned, animals were fistulated when they were 12 months old. After the surgery by skilled veterinarian, calves suffered from temporal loss of appetite, but no longer than 2 or 3 days, and no apparent adverse effects of cannulation were observed as similar to previous report (Kristensen et al., 2010; Technical note: Ruminal cannulation technique in young Holstein calves: Effects of cannulation on feed intake, body weight gain, and ruminal development at six weeks of age) although there is differences in age (6 weeks vs. 12 months of age) and breed (Holstein vs. Japanese Black cattle). Furthermore, no sign of obvious illness and antibiotic use was observed during the experimental period as mentioned in the original manuscript L363-365, which also are added in the Result section L220-221. Therefore, the effect of surgery was minimized in the present study, and no additional dietary adaptation was needed because Early fattening stage diet was fed before 2 months before surgery (10 months of age). Even if considering at least one month for recovery in the best case scenario and another 3 weeks for diet adaption, we considered that calves were fully adapted to Early stage diet during 13 to 14 months of age.

 Also, minor considerations should be the forage-to-concentrate ratio that was modified throughout the experiment, being 26:74 during Early stage, 13:87 during the Middle stage, and 14:86 during Late stage, which might have affected the rumen environment and pH values. Also, the diet composition should be presented, and the methods used for chemical analysis. Vitamin A concentration should also be included in the Table 1, as you discussed hypovitaminosis in the Introduction section.

AU: We would like to appreciate for your comment. As you mentioned, the diet composition was modified throughout the experiment period, and it affected the rumen environment and pH values. During the fattening period, dietary modification according to different fattening stage is a common practice in Japanese Black cattle (original manuscript L343-345), and we followed general agreement in Japan (research and commercial farm) as introduced previously (Oka et al., 1998; Ogata et al., 2019). Furthermore, the present experiment presupposed dietary modification during the entire fattening stages to explore long-term changes in ruminal pH and fermentation, as well as their consequences with respect to the rumen bacterial community (original manuscript L345-347). However, we would like to apology for our mistake that the forage-to-concentrate ratio in the original manuscript was based on the intake amount, and it is revised accordingly in the revised manuscript L89-90 as “The forage-to-concentrate ratio was gradually decreased from 38:62 to 22:78, 13:87 to 10:90 and 8:92 to 7:93 during the Early, Middle, and Late stages, respectively”. In addition, chemical composition of the diets was analyzed according to the official method analysis of the Association of Official Analytical Chemists (AOAC) that registered in the Official Method Feed Analysis of Japan (MAFF, 2008) as we mentioned in the revised manuscript L98-101. Regarding the vitamin A concentration in the diet, we agree with your comment that it should be included in the Table 1. However, because vitamin A concentration is not directly related to our experiment, we did not measure dietary vitamin A concentration, and it can be predicted by peripheral blood concentration. 

 Finally, the lack of mechanical lysis of rumen content is troublesome and data interpretation and extrapolation should be made carefully. Several studies have shown that disruption of bacteria with tough cell walls is more efficient with a mechanical approach than by an enzyme-based protocol. Furthermore, extraction methods have important implications on the results, and studies using different extraction procedures should not be compared. For that, the conclusion needs to be re-worked.

AU: We would like to appreciate for your comment, and partly agree with your comment about mechanical lysis of rumen content. As you mentioned, DNA extraction methods and sampling techniques may affect the rumen microbial community structure (Henderson et al., 2013; Effect of DNA extraction methods and sampling techniques on the apparent structure of cow and sheep rumen microbial communities). In our study, total bacterial DNA was extracted as described previously Morita et al. (2007; An improved DNA isolation method for metagenomics analysis of the microbial flora of the human intestine) with minor modifications. Morita et al. (2007) reported that the improved DNA extraction method using lysozyme, proteinase K, and achromopeptidase gave stable and high-level lysis (>90%) for all the human fecal samples compared to the reference method (13.3-84.6%) and QIAamp DNA stool mini kit (38.8-69.2%). In addition, our study aimed to explore changes in the rumen fluid bacterial community after filtering through two layers of cheesecloth, and nearly no rumen contents are included in the fluid sample, possibly suggesting mechanical lysis of rumen contents has not much meaning in the present study. Unfortunately, to the best of our knowledge, we could not find studies that used enzymatic DNA extraction method in our field. Therefore, our DNA extraction method may reflect nearly true genomic information in the rumen microbial flora, while there are not many citable references in this kind of study.

 Despite these considerations, the paper is well-prepared. Studies exploring the rumen microbiome and metabolic disorders are needed. If necessary, I would be available to look at the revised version.

AU: Authors would like to gratefully appreciate Reviewer 1 for your helpful comments and suggestions. Regarding studies exploring the rumen microbiome and metabolic disorder, a sentence about health problems induced by SARA is added in the revised manuscript L360-362 as “The occurrence of SARA may cause various health problems in cattle, such as feed intake depression, reduced fiber digestion, milk fat depression, diarrhea, laminitis, liver abscesses, increased production of bacterial endotoxins, and inflammation (Plaizier et al., 2008)”. 

Specific comments

 Please, verify financial disclosure guidelines and amend if appropriate. According to PLOS ONE guidelines, Funded studies should have statements with the following details: Initials of the authors who received each award; Grant numbers awarded to each author; The full name of each funder; URL of each funder website; and whether the sponsors or funders play any role in the study design, data collection and analysis, decision to publish, or preparation of the manuscript.

AU: We would like to appreciate for your comment. However, we do not have any grant information in this study. Therefore, we did not declare grant information in the manuscript. 

P02L26: what ‘specialized high-concentrate diets’ means?

AU: In the present study, high-concentrate diets were not commercially available merchandise, but specially designed for the experimental cattle based on the Japanese Feeding Standard for Beef cattle. The relevant description is revised accordingly as “Specially designed high-concentrate diet” in the revised manuscript L26 and L86.

P02L29: the term ‘were collected on day 4 of pH measurement’ is confusing.

AU: In the present study, ruminal pH was measured during the final 7 days (days 1-7) of each fattening stage as described in the manuscript L101. The sentence “were collected on day 4 of pH measurement” is revised more in detail as “were collected on day 4 (fourth day during the final 7 days of pH measurement) during the ~” in the revised manuscript L29-30, L113-114, and L186-187.

P02L45-46: I respectfully disagree with the first sentence of the Introduction section, as pasture based diet can also promote growth, productivity, and high-quality meat or milk.

AU: We would like to appreciate for your comment on our thought. We completely agree with your comment, and people, including us, easily overlook the importance of forage and the relevant things that the ruminants live on pasture. Recently, however, high-concentrate based diet is well known for high-energy property and used broadly because of a human’s purpose in the world. Therefore, the sentence is revised accordingly as “A high-grain based diet is essential for beef and dairy cattle, to maximize growth, productivity, and high-quality meat or milk.” in the revised manuscript L46-47.

P04L48-50: there are parts of the manuscript which are confusing, such as the phrase ‘As a result, ruminal pH decreases; subacute ruminal acidosis (SARA) and ruminal acidosis (RA) are defined by ruminal pHs of ≤ 5.6 and below, respectively (Nagaraja and Titgemeyer, 2007).’. For example, I would suggest modifying to ‘ruminal pHs values of ≤ 5.6’.

AU: We would like to thank you for your comment. The relevant sentence is revised according to your comment in the revised manuscript L49-51. 

P05L69-71: this phrase should be re-worked as you did not evaluate the ‘effect of ruminal pH, bacterial community and fermentation on the fattening of 10-month-old Japanese Black beef cattle’. Rather, this work characterized ruminal pH, bacterial community and fermentation characteristics on different ages of Japanese Black beef cattle.

AU: We would like to appreciate for your comment. The relevant sentence is revised accordingly in the revised manuscript L70-71 as “we explored the effects of the ruminal pH, bacterial community and fermentation characteristics on different ages of Japanese Black beef cattle”.

P05L71: ‘this findings’ is incorrect.

AU: We would like to apology for our careless mistake. This sentence is revised accordingly in the revised manuscript L72 as “these findings”. 

P06L82: what percentage of refusals?

AU: We would like to thank you for your comment. During the Early stage, a fixed amount of concentrate and forage diets were offered, and feed refusal rate of concentrate and forage diets were 12.6% and 12.2%, respectively as revised in the revised manuscript L88-89. However, diets were fed ad libitum during the Middle and Late stages, and the percentage of refusal was not accessed during these stages. 

P06L83: why 10–14, 15–22, and 23–30 months of age were selected as different stages? It seems arbitrary. This should be addressed.

AU: As we mentioned above, there is general agreement in Japanese Black cattle fattening stage, which included Early, Middle, and Late stages (10–14, 15–22, and 23–30 months of age, respectively). Although specific months of age is not documented, many research groups and commercial farms followed this guideline as reported previously (Oka et al., 1998; Ogata et al., 2019; Maeda et al., 2019; Effect of feeding wood kraft pulp on the growth performance, feeding digestibility, blood components, and rumen fermentation in Japanese Black fattening steers). Therefore, we followed general agreement in Japan as introduced previously (Oka et al., 1998; Ogata et al., 2019), and this information is added in the revised manuscript L84-86 as “The fattening stages included Early, Middle, and Late stages (10–14, 15–22, and 23–30 months of age, respectively) according to general agreement in Japan (Oka et al., 1998; Ogata et al., 2019)”.

P06L89: the rationale to feed concentrate after 1 hour of forage should be addressed. Also, how concentrate availability was ensured?

AU: Generally, cattle consumes concentrate diet first, and then, access to forage diet latter. Under our high-grain diet feeding condition, the cattle may consume excessive amount of concentrate diet and suffer from severe depression of ruminal pH, such as acute ruminal acidosis. Therefore, to prevent possible disorder, we decided to offer forage diet in advance to concentrate diet. The relevant information is added in the revised manuscript L94-96 as “concentrate diet was supplied 1 h after a forage diet feeding to maximize forage diet intake and to prevent excessive consumption of concentrate diet during the Early stage”. In addition, diets were offered individually to each cattle, and daily intake amounts of concentrate and forage were also recorded individually. Therefore, there was no problem in concentrate diet availability during the entire study period. 

P06L93: the term ‘sufficient rate’ is dubious. Use other term.

AU: We would like to apology for our dubious term. At first, we intended to describe how much offered feed can cover an expected daily weight gain in the experimental cattle. However, we choose inappropriate term in the relevant description as “sufficient rate”, and thus, we revise this description accordingly as “Nutrient adequacy rate” in the revised Table 1. Furthermore, supplementary description is added in the footnote as “Nutrient adequacy rate was based on the nutrient requirement of Japanese Feeding Standard for Beef Cattle (National Agriculture and Food Research Organization (NARO), 2009), with an expected daily weight gain of 0.8, 0.65, and 0.7 kg during the Early, Middle, and Late stages, respectively”.

P06L93: please, refer the Japanese feeding standard.

AU: We would like to apology for our careless mistake. The reference is added accordingly in the revised manuscript L101-102 as “The adequacy rate of diet was calculated based on the nutrient requirement of Japanese Feeding Standard for Beef cattle (NARO, 2009)”

P08L104: the phrase ‘Rumen fluid samples were collected on day 4 of the pH measurements during the Early…’ is confusing and should be re-worked.

AU: As we mentioned above, this sentence “were collected on day 4 of pH measurement” is also revised more in detail as “were collected on day 4 (fourth day during the final 7 days of pH measurement) during the ~” in the revised manuscript L113-114.

P09L122: there are two references of Kim et al. 2016, which should be differentiated according to PLOS ONE manuscript preparation guidelines.

AU: We would like to apology for our careless mistake. The references are distinguished accordingly (Kim et al., 2016a; Front Microbiol, and Kim et al., 2016b; Physiol Genomics) in the revised manuscript.

P09L128: what was the ratio of phenol/chloroform/isoamyl alcohols?

AU: The ratio of phenol/chloroform/isoamyl alcohols was 25:24:1, and the relevant information is added in the revised manuscript L137-138 as “with phenol/chloroform/isoamyl alcohol (25:24:1) (Wako Pure Chemical Industries Ltd.)”.

P09L134: authors need to clarify whether pyrosequencing was the used approach. Furthermore, this should be addressed throughout the manuscript.

AU: We would like to appreciate for your comment. In the present study, pyrosequencing approach was performed followed by manufacturer’s instruction (16S Metagenomic Sequencing Library Preparation; Preparing 16S Robosimal RNA Gene Amplicons for the Illumina Miseq System). Therefore, the information is added in the revised manuscript L144-145 as “Sequencing libraries preparation was performed according to the Illumina 16S Metagenomic Sequencing Library preparation guide (2013)”.

P11L175: the phrase ‘Rumen fluid samples were collected on day 4 of the pH measurements during the Early…’ is confusing and should be re-worked.

AU: As we mentioned above, this sentence “were collected on day 4 of pH measurement” is also revised more in detail as “were collected on day 4 (fourth day during the final 7 days of pH measurement) during the ~” in the revised manuscript L186-187. 

Table 2: please, use ‘ab’ to indicate similarities between stages. For example, the minimum pH values should be presented as ‘5.43a, 5.30ab, and 4.98b’ for Early, Middle, and Late stages, respectively. This should be considered for other tables.

AU: We would like to appreciate for your helpful comment, and to apology for our careless mistake. As you mentioned, superscript ‘ab’ is added throughout the tables. 

Table 3: data on acetic, propionic and butyric acids should be discussed in the text.

AU: We would like to thank you for your suggestion. In the present study, the proportions of these acids were not different throughout the fattening period, while they showed general feature of high-grain diet feeding in Holstein cattle as reported elsewhere (Khafipour et al., 2009; Nagata et al., 2018). Therefore, the data on acetic, propionic, and butyric acids are discussed in the Discussion section L383-385 as “Furthermore, decrease and increase in the proportions of acetic and propionic acids, respectively, were consistent with general feature of high-grain diet feeding in Holstein cattle (Khafipour et al., 2009; Nagata et al., 2018)”.

P26L349: the effect of feed intake reduction in the Late stage should be discussed. Did animals decrease feed intake, and reduced concentrate intake by 20% to mitigate health problems?

AU: Throughout the study period, animals did not show any health problems, such as high-temperature, dehydration, and diarrhea. In addition, there was no abnormal or critical (within physiological range) changes in blood metabolites during the fattening stages, and thus, we could not suggest any plausible answer to the reason. In the revised manuscript, these information are clarified in the Result (L220-221) and Discussion (L363-365) sections, and decreased feed intake is discussed in the Discussion (L367-368). 

P27L385: the discussion on reduction of ruminal bacterial richness but not bacterial diversity, should not be limited to one study. Please expand this discussion using other studies that corroborate with you study, other which does not, as it is central to your work. I would also consider bacterial and sequencing limitations, as bead beating was not used.

AU: We would like to appreciate for your helpful comment. We also agree with your comment that discussion should be expanded using other studies that corroborate with our study. In the original manuscript of the Discussion section L396-399, we mentioned that “bacterial diversity and richness indices, correlations of pH parameters with bacterial diversity and richness were generally positive; low ruminal pH leads to low bacterial diversity and richness (Khafipour et al., 2009; Mao et al., 2013; Plaizier et al., 2017; Nagata et al., 2018)”. However, to our best knowledge, this is the first study demonstrating the relationship between the long-term high-grain diet feeding and bacterial diversity or richness. Therefore, we clarify this in the revised manuscript L405-407 as “To best our knowledge, this is the first study demonstrating the relationship between the long-term high-grain diet feeding and bacterial diversity or richness, and suggests that ~”.

P28L401-405: the different core microbiota observed in your study compared to the literature was expected. A diet enriched in concentrate was used, having greater levels compared to dairy cattle diets. Furthermore, dairy cattle is anatomically different to beef cattle, which has implication of passage rate for example, and passage rate has huge influence on the rumen microbiota. Finally, generally beef cattle is fed to maximize productivity and to produce beef with quality. On the other hand, dairy cattle are fed to maximize productivity, but considering health and reproductivity. Thus, nutrient requirements and managements are different.

AU: We would like to appreciate for your comment. Regarding the relevant differences in core microbiota in our study to Holstein cattle, we also agree with your comment. Therefore, the manuscript is revised accordingly to include your comment as “~ the genus Prevotella was generally the most predominant in the rumen bacterial community of Holstein cattle (Mao et al., 2013; Golder et al., 2014; Kim et al., 2016a; Nagata et al., 2018). This is because different breeds of cattle may have different feed passage rate through the digestive tract due to different eating and rumination behaviors (Aikman et al., 2008), and Holstein cows are fed high-grain based diet to maximize productivity, but considering health and reproductivity (Roche, 2006), compared with those fed to maximize productivity and to produce beef with quality (Ogata et al., 2019).” in the revised manuscript L425-431.

 

Reviewer #2: Line Comment

AU: Authors would like to thank Reviewer 2 for your helpful comments and suggestions. We have done our best to address all the issues raised by Reviewer 2 very carefully in this new revision, which we believe has improved the quality of the paper further. 

29 It this total VFA concentration?

AU: We would like to apology for our ambiguous description. The description is revised accordingly as “total VFA production” in the revised manuscript L41.

72 Perhaps “understand” could be changed to “the understanding of”.

AU: We would like to appreciate for your suggestion. We revise the expression according to your suggestion as “the understanding of” in the revised manuscript L73-74.

81, 374 It is noted that the cattle were rumen-fistulated at 12-mnths of age, during the Early stage of the trial. Please speculate whether this would have influenced the results obtained during that phase. It was also noted that fistualtion was said to be done at 10-months of age on line 374. If cattle were fistulated at 10 months, no discussion of affects on the animals is necessary.

AU: We would like to thank you for your comment. As you mentioned as a response to Reviewer 1, no apparent adverse effects of cannulation were observed in the present study. After the surgery at 12 months of age by skilled veterinarian, calves suffered from temporal loss of appetite, but no longer than 2 or 3 days, and no apparent adverse effects of cannulation were observed as similar to previous report (Kristensen et al., 2010; Technical note: Ruminal cannulation technique in young Holstein calves: Effects of cannulation on feed intake, body weight gain, and ruminal development at six weeks of age) although there is differences in age (6 weeks vs. 12 months of age) and breed (Holstein vs. Japanese Black cattle). Furthermore, no sign of obvious illness and antibiotic use was observed during the experimental period as mentioned in the original manuscript L363-365, which also are added in the Result section L220-221. Therefore, we believe that the effect of surgery was minimized in the present study.

85, 88, 90, etc. Suggest using either “roughage” or “forage” for that portion of the diet.

AU: We would like to appreciate for your suggestion. “roughage” is revised to “forage” throughout the revised manuscript. 

93, Table 1 What is “sufficient rate”? Is this a requirement? If it is a requirement, why are the units in “%”? Are the dairy intakes in kg of DM? Please clarify. Also, please supply a citation for the Japanese feeding standard.

AU: We would like to apology for our dubious term. As we responded to Reviewer 1, we intended to describe how much offered feed can cover an expected daily weight gain in the experimental cattle. However, we choose inappropriate term in the relevant description as “sufficient rate”, and thus, we revise this description accordingly as “Nutrient adequacy rate” in the revised Table 1. Furthermore, supplementary description is added in the footnote as “Nutrient adequacy rate was based on the nutrient requirement of Japanese Feeding Standard for Beef Cattle (NARO, 2009), with an expected daily weight gain of 0.8, 0.65, and 0.7 kg during the Early, Middle, and Late stages, respectively”. Regarding the Japanese Feeding standard, reference is added accordingly in the revised manuscript L101-102 as “The adequacy rate of diet was calculated based on the nutrient requirement of Japanese Feeding Standard for Beef cattle (NARO, 2009).”

117 Was the supernatant or the pellet analyzed for LPS activity? I ask because the LPS is presumably associated with the microbes and likely would be largely with the 11,000 x g pellet.

AU: We would like to thank you for your comment. In the present study, rumen fluid samples were centrifuged and supernatant and pellet were separated. Then, LPS activity was assayed using supernatant to minimize interruption of gram-negative bacterial cell membrane. Therefore, we clarify that supernatant LPS activity was assayed using a kinetic Limulus amebocyte lysate assay as “~ and supernatant LPS activity was assayed using a kinetic Limulus amebocyte lysate assay ~” in the revised manuscript L126-127.

213-217 Again, please clarify what is meant by “rates”.

AU: We would like to apology for our undescriptive description. As we mentioned above, “sufficient rate” is revised to “nutrient adequacy rate” in the Table 1, and thus, the relevant description “rate” is revised in detail as “Nutrient adequacy rates” in the revised manuscript L225.

Table 2 Why are there no superscripts on several of the values for the Middle treatment (as is seen in Table 4)? Normally, one would expect that, if those values were not different from Early and/or Late, they would share the superscript with the Early and/or Late. Sometimes the Middle values are intermediate, or even greater than, either Early and Late.

AU: We would like to appreciate for your helpful comment, and to apology for our careless mistake. As you and Reviewer 1 mentioned, superscript ‘ab’ is added throughout the tables. 

Table 3 Why are there no superscripts for A/P ration for the Late treatment?

AU: We would like to appreciate for your helpful comment, and to apology for our careless mistake. As we mentioned above, superscript ‘ab’ is added throughout the tables. 

383-385 Do you mean to say that the risk of SARA during the Early stage was low, because pH was actually higher during this stage?

AU: In the present study, ruminal pH during the Early fattening stage was higher than other stages. Although the duration of time where pH < 5.6 was not long to diagnose as SARA during the Early stage, ruminal pH value was already low to induce low bacterial diversity from the beginning of the fattening stage when compared with previous study (Nagata et al., 2018). Also, the depression was getting more severe during the latter period, resulting in reduced rumen bacterial richness but not bacterial diversity. Therefore, our conclusion was that the risk of SARA during the Early stage was “high”, because pH was actually “low” during the Early stage although we did not diagnose SARA based on the duration of time in the present study. 

Additional comment 

Normally, one would expect to see a table of diet composition, showing the feedstuffs used to construct the 3 diets fed in this trial. The authors might consider adding a table containing this information to the paper.

AU: We would like to thank you for your suggestion. During the three different fattening period, the cattle were fed simply two kind of diets, concentrate (specially designed for the experimental cattle) and rice straw. Therefore, forage in the Table 1 is revised in detail as “Rice straw” in the revised manuscript, and we consider that this can explain the feedstuffs used to construct the 3 diets in this study.

---

## [Decision Letter · Decision Letter 1]

23 Oct 2019

PONE-D-19-20947R1

Long-term high-grain diet altered the ruminal pH, fermentation, and composition and functions of the rumen bacterial community, leading to enhanced lactic acid production in Japanese Black beef cattle during fattening

PLOS ONE

Dear Dr Sato, 

Thank you for submitting your manuscript to PLOS ONE. After careful consideration, we feel that it has merit but does not fully meet PLOS ONE’s publication criteria as it currently stands. Therefore, we invite you to submit a revised version of the manuscript that addresses the points raised during the review process.

My decision is based on the comments made by one of the reviewers that have made several more suggestions to improve the manuscript. I concur with this view and I will be glad to receive the revised manuscript soon.

We would appreciate receiving your revised manuscript by November 10th. To enhance the reproducibility of your results, we recommend that if applicable you deposit your laboratory protocols in protocols.io, where a protocol can be assigned its own identifier (DOI) such that it can be cited independently in the future. For instructions see: http://journals.plos.org/plosone/s/submission-guidelines#loc-laboratory-protocols

We look forward to receiving your revised manuscript.

Kind regards,

Marcio de Souza Duarte

Academic Editor

PLOS ONE

Reviewers' comments:

Reviewer's Responses to Questions

**Comments to the Author**

1. If the authors have adequately addressed your comments raised in a previous round of review and you feel that this manuscript is now acceptable for publication, you may indicate that here to bypass the “Comments to the Author” section, enter your conflict of interest statement in the “Confidential to Editor” section, and submit your "Accept" recommendation.

Reviewer #1: All comments have been addressed

Reviewer #2: All comments have been addressed

2. Is the manuscript technically sound, and do the data support the conclusions?

Reviewer #1: Yes

Reviewer #2: Yes

3. Has the statistical analysis been performed appropriately and rigorously? 

Reviewer #1: Yes

Reviewer #2: Yes

4. Have the authors made all data underlying the findings in their manuscript fully available?

Reviewer #1: Yes

Reviewer #2: Yes

5. Is the manuscript presented in an intelligible fashion and written in standard English?

Reviewer #1: Yes

Reviewer #2: Yes

6. Review Comments to the Author

Reviewer #1: Comments to the Author

I appreciate the explanations and revisions of the authors and believe, I can provide better comments and suggestions.

L83: please provide the body weight of animals prior to the experiment;

L86-91: the concentrate composition is still missing. There are several considerations about animal performance, and the reluctancy to include feed composition troubles me. I understand that authors might have another paper with performance and other data and if so, please discuss accordingly in this manuscript. I also understand the performance was not the objective of this manuscript. However, intake and performance are central to discuss rumen microbiome modulation. For example, according to the provided body weights, animals gained 122 kg from Early to Middle, with an average daily gain of 0.51, which is 21.79 % lower from the expected daily weight gain. Furthermore, animals had an average daily gain of 0.63 kg and the estimation was 0.7 kg, thus animals performed 10.12 % less than what was expected for the Late stage. Moreover, feed intake decreased by 18% (Table 1) from Middle to Late stages. Yet animals had a gain in average daily weight. What is the rationale of this phenomenon?

L85-88: please provide the concentrate composition rather the phrase ‘specially designed high-concentrate’;

L89: authors included new values for forage-to-concentrate ratio. My consideration about forage-to-concentrate ratio was that it was ‘modified throughout the experiment, being 26:74 during Early stage, 13:87 during the Middle stage, and 14:86 during Late stage, which might have affected the rumen environment and pH values’. Forage-to-concentrate ratio was properly presented. However, the increase percentage of concentrate in the diet and how it affected the rumen microbiome was not satisfactorily discussed in the previous version of the manuscript. I would suggest author to keep forage-to-concentrate ration as it was in the first version, or clarify if the new value is the recommend values according to practices in Japan;

L89-91: I would consider being more precise in the description of gradually decreased. Was it a weekly adjustment for example?

L91: the body weight should be 562 instead of 561. According to data on table 1, the weight was 561.8, thus it should be rounded up accordingly;

L96-97: do authors collected feed intake for the whole trial, or only during the sampling weeks? If that was the case, please address possible limitations of this approach in the discussion section. For example, animals could have suffered acute feed intake by the end of the animal trial, which help explain intake and performance data;

Table 1: please indicate if data for daily intake amount is based on organic or dry matter.

L143;160;161: as I understand, you used the Illumina platform, which is based on reversible dye-terminator instead of pyrosequencing. Usually the pyrosequencing was employed using the Roche 454 sequencing platform. Please, address this accordingly;

L364-365: since body temperature, dehydration and diarrhea were considered clinical signs, please include how these were monitored in the Material and Methods sections;

L366-368: in the Late stage animals experienced rumen pH value under 5.6 for almost 11.5 hours, decreased VFA production, increased lactic acid, LPS, and aspartate transaminase concentration, which suggests SARA and decreased feed intake. However, check your data on feed intake and forage-to-concentrate ratio;

Table 5: Please, verify if differences between stages are correct for the OUT8. Should Late and Middle stage be similar?

Reviewer #2: (No Response)

7. PLOS authors have the option to publish the peer review history of their article (what does this mean?). If published, this will include your full peer review and any attached files.

Reviewer #1: No

Reviewer #2: No

---

## [Author Response · Author response to Decision Letter 1]

1 Nov 2019

PONE-D-19-20947R1

Long-term high-grain diet altered the ruminal pH, fermentation, and composition and functions of the rumen bacterial community, leading to enhanced lactic acid production in Japanese Black beef cattle during fattening

Dear Editor and Reviewers

 Authors would like to appreciate Editor and Reviewers for your helpful comments and suggestions. We have done our best to address all the issues raised by Reviewer #1 very carefully in this second revision, which we believe has improved the quality of the paper further. To facilitate the reviewing process, we have highlighted all changes done by Authors (Yellow) in the revised manuscript. We have responded to every comment done by the Reviewer #1 below, and also have indicated the changes made with respective new lines. 

6. Review Comments to the Author

Reviewer #1: Comments to the Author

I appreciate the explanations and revisions of the authors and believe, I can provide better comments and suggestions.

AU: Authors would like to thank Reviewer #1 for your helpful comments and suggestions. We have done our best to address all the issues raised by you very carefully in this new revision, which we believe has improved the quality of the paper further. 

L83: please provide the body weight of animals prior to the experiment;

AU: We would like to appreciate for your comment. The body weight of animals prior to the experiment (10 months of age) is added in the revised manuscript L92-94 as “The mean ± SE body weight of the cattle was 335 ± 4.4, 439 ± 7.6, 562 ± 11.6, and 712 ± 18.5 kg on prior to the experiment (10 months of age), and Early (14 months of age), Middle (21 months of age), and Late (29 months of age) fattening stage sampling days, respectively.”.

L86-91: the concentrate composition is still missing. There are several considerations about animal performance, and the reluctancy to include feed composition troubles me. I understand that authors might have another paper with performance and other data and if so, please discuss accordingly in this manuscript. I also understand the performance was not the objective of this manuscript. However, intake and performance are central to discuss rumen microbiome modulation. For example, according to the provided body weights, animals gained 122 kg from Early to Middle, with an average daily gain of 0.51, which is 21.79 % lower from the expected daily weight gain. Furthermore, animals had an average daily gain of 0.63 kg and the estimation was 0.7 kg, thus animals performed 10.12 % less than what was expected for the Late stage. Moreover, feed intake decreased by 18% (Table 1) from Middle to Late stages. Yet animals had a gain in average daily weight. What is the rationale of this phenomenon?

AU: We would like to apology for our missing data. The concentrate composition is added in the revised manuscript L87-90 as “The concentrate diet was composed of barely, steam-flaked corn, wheat bran, and soybean meal and contains 71.2% total digestible nutrient (TDN) and 15.7% crude protein (CP), 72.2% TDN and 13.9% CP, and 72.8% TDN and 12.0% CP during the Early, Middle, and Late stage, respectively.”

Regarding the animal performance, we partly agree with your opinion that intake and performance are central to discuss rumen microbiome modulation. In the present study, actual daily gain of cattle was observed only 2 weeks of period at each stage (from 1 week before to 1 week after sample collection day), and average daily gain was 0.75 (93.8% than calculated amount for daily gain), 0.69 (106.2%), and 0.46 (65.7%) kg/day during the Early, Middle, and Late stages, respectively. In addition, average daily gain during the entire experimental period was 0.63 kg/day in the present study. However, as you mentioned, the performance was not the main objective of the present study. Furthermore, the Japanese Feeding Standard for Beef cattle aimed to increase intramuscular fat accumulation (highly marbled meat) as discussed in the Introduction and Discussion sections, which may not fit the general farming object in other countries that maximize growth performance of beef cattle. We also agree with the fact that feed intake is undoubtedly important to meet energy requirement for production. However, other factors, such as peripheral hormones, environment, and genetic strain, also significantly affect the animal performance. As an example, peripheral blood vitamin A level is known for modulating intramuscular fat deposition, and the fattening cattle are generally fed high-grain, low vitamin A-containing diets to induce greater intramuscular fat deposition, leading to highly marbled meat during the fattening period (Oka et al., 1998) as introduced in the Introduction (L62-66) and Discussion (L355-357) sections. Therefore, we considered that descriptions in our manuscript is sufficient to concisely discuss and suggest our study aim. Although our description is somewhat unclear to elucidate the rationale of the phenomenon, we alternatively added our opinion on the dietary intake in the revised manuscript L365-367 as “Dietary intake amounts were highest during the Middle stage and lowest during the Late stage as an adaptation to long-term high-grain diet feeding or response to significantly lowered ruminal pH during the latter fattening stage”. 

L85-88: please provide the concentrate composition rather the phrase ‘specially designed high-concentrate’;

AU: We would like to apology for our missing data. As we mentioned above, concentrate composition is added in the revised manuscript L87-90. Please see the revised manuscript. 

L89: authors included new values for forage-to-concentrate ratio. My consideration about forage-to-concentrate ratio was that it was ‘modified throughout the experiment, being 26:74 during Early stage, 13:87 during the Middle stage, and 14:86 during Late stage, which might have affected the rumen environment and pH values’. Forage-to-concentrate ratio was properly presented. However, the increase percentage of concentrate in the diet and how it affected the rumen microbiome was not satisfactorily discussed in the previous version of the manuscript. I would suggest author to keep forage-to-concentrate ration as it was in the first version, or clarify if the new value is the recommend values according to practices in Japan;

AU: We would like to thank you for your valuable suggestion, and the relevant description about forage-to-concentrate ratio is reversed as it was in the original manuscript. Regarding the increase percentage of concentrate in the diet and how it affected the rumen microbiome, the dietary composition (e.g. forage-to-concentrate ratio) was changed monthly during the experiment period as revised in the previous revision despite it is reversed to the original description. However, rumen fluid sample was collected only once at the end of each fattening period, and sample collection is not accord with the dietary compositional changes (monthly dietary change vs. fattening periodic sample collection). As a result of feeding management and as we presented in the manuscript, total VFA concentration decreased, and lactic acid concentration increased during the Late stage as the adaptation or response to long-term high-grain diet feeding and long-term high-grain diet compositional change. Although we fully agree with your valuable suggestion on further discussion on the dietary change and rumen microbiome, we believe that our data connected long-term fattening diet feeding to ruminal pH, rumen fermentation, bacterial community composition and structure, blood metabolites, their relationships, and their consequences to leading enhanced lactic acid production are presented accordingly in the present study.

L89-91: I would consider being more precise in the description of gradually decreased. Was it a weekly adjustment for example? 

AU: We would like to appreciate for your comment. However, as you mentioned above, the relevant description is reversed as it in the original manuscript. Please see the revised manuscript L91-92.

L91: the body weight should be 562 instead of 561. According to data on table 1, the weight was 561.8, thus it should be rounded up accordingly;

AU: We would like to apology for our careless mistake. The body weight is revised accordingly as 562 in the revised manuscript L92.

L96-97: do authors collected feed intake for the whole trial, or only during the sampling weeks? If that was the case, please address possible limitations of this approach in the discussion section. For example, animals could have suffered acute feed intake by the end of the animal trial, which help explain intake and performance data;

AU: We would like to appreciate for your helpful comment. In the present study, feed intake amount was collected daily throughout the experiment period. In addition, we checked that there was no acute increase or decrease in feed intake amount throughout the study period. Therefore, we added the information in the revised manuscript L364 as “no clinical sign of abnormal body condition, such as high body temperature, acute feed intake, dehydration ~”.

Table 1: please indicate if data for daily intake amount is based on organic or dry matter.

AU: We would like to apology for our undetailed description. In the Table 1, daily intake amount was based on the organic matter, and it is added in the footnote of the Table 1. 

L143;160;161: as I understand, you used the Illumina platform, which is based on reversible dye-terminator instead of pyrosequencing. Usually the pyrosequencing was employed using the Roche 454 sequencing platform. Please, address this accordingly;

AU: We would like to appreciate for your comment. The description is revised accordingly in the revised manuscript L145 (Library preparation and DNA sequencing), 161 (Sequencing data analyses), and 162 (All sequencing reads were~).

L364-365: since body temperature, dehydration and diarrhea were considered clinical signs, please include how these were monitored in the Material and Methods sections;

AU: We would like to appreciate for your suggestion. The relevant information is added in the revised manuscript L97-98 as “Abnormalities of body condition (body temperature, appetite, hydration, and defecation) were observed daily throughout the study period”.

L366-368: in the Late stage animals experienced rumen pH value under 5.6 for almost 11.5 hours, decreased VFA production, increased lactic acid, LPS, and aspartate transaminase concentration, which suggests SARA and decreased feed intake. However, check your data on feed intake and forage-to-concentrate ratio;

AU: We would like to appreciate for your comment. In the original manuscript, we interpreted the relevant results with caution to avoid any overspeculation beyond our study result and aim. Therefore, as you mentioned above, we carefully suggest the reasons for decreased feed intake during the Late stage in the revised manuscript L365-367 as “Dietary intake amounts were highest during the Middle stage and lowest during the Late stage as an adaptation to long-term high-grain diet feeding or response to significantly lowered ruminal pH during the latter fattening stage”.

Table 5: Please, verify if differences between stages are correct for the OTU8. Should Late and Middle stage be similar?

AU: We would like to appreciate for your comment and apology for our careless mistake on the statistical analysis. As you mentioned, we checked all the validity of data, and found statistical mistakes only in the Table 5. Therefore, we revised the relevant descriptions in the Result (L311-324) and superscripts in the Table 5. However, these revisions have any influence on other part of the manuscript.

---

## [Editor Report · Decision Letter 2]

6 Nov 2019

Long-term high-grain diet altered the ruminal pH, fermentation, and composition and functions of the rumen bacterial community, leading to enhanced lactic acid production in Japanese Black beef cattle during fattening

PONE-D-19-20947R2

Dear Dr. Sato,

We are pleased to inform you that your manuscript has been judged scientifically suitable for publication and will be formally accepted for publication once it complies with all outstanding technical requirements.

With kind regards,

Marcio de Souza Duarte

Academic Editor

PLOS ONE
---

## [Editor Report · Acceptance letter]

13 Nov 2019

PONE-D-19-20947R2 

Long-term high-grain diet altered the ruminal pH, fermentation, and composition and functions of the rumen bacterial community, leading to enhanced lactic acid production in Japanese Black beef cattle during fattening 

Dear Dr. Sato:

I am pleased to inform you that your manuscript has been deemed suitable for publication in PLOS ONE. Congratulations! Your manuscript is now with our production department. 

With kind regards,

on behalf of

Dr. Marcio de Souza Duarte 

Academic Editor

PLOS ONE